# Odd skipped-related 1 identifies a population of embryonic fibro-adipogenic progenitors regulating myogenesis during limb development

Pedro Vallecillo-García [1,2], Mickael Orgeur[1,2,3,4], Sophie vom Hofe-Schneider [1], Jürgen Stumm[1,2], Verena Kappert[1,2], Daniel M. Ibrahim [2], Stefan T. Börno[2], Shinichiro Hayashi[5,6], Frédéric Relaix[5,6], Katrin Hildebrandt[7], Gerhard Sengle[7], Manuel Koch [7], Bernd Timmermann[2], Giovanna Marazzi[8,9], David A. Sassoon[8,9], Delphine Duprez [3,4] & Sigmar Stricker [1,2]

Fibro-adipogenic progenitors (FAPs) are an interstitial cell population in adult skeletal muscle that support muscle regeneration. During development, interstitial muscle connective tissue (MCT) cells support proper muscle patterning, however the underlying molecular mechanisms are not well understood and it remains unclear whether adult FAPs and embryonic MCT cells share a common lineage. We show here that mouse embryonic limb MCT cells expressing the transcription factor Osr1, differentiate into fibrogenic and adipogenic cells in vivo and in vitro defining an embryonic FAP-like population. Genetic lineage tracing shows that developmental Osr1+ cells give rise to a subset of adult FAPs. Loss of *Osr1* function leads to a reduction of myogenic progenitor proliferation and survival resulting in limb muscle patterning defects. Transcriptome and functional analyses reveal that Osr1+ cells provide a critical pro-myogenic niche via the production of MCT specific extracellular matrix components and secreted signaling factors.

[1] Institute for Chemistry and Biochemistry, Freie Universität Berlin, D-14195 Berlin, Germany. [2] Max Planck Institute for Molecular Genetics, D-14195 Berlin, Germany. [3] Sorbonne Universités, UPMC Univ Paris 06, CNRS UMR7622, IBPS-Developmental Biology Laboratory, F-75005 Paris, France. [4] Inserm U1156, IBPS-Developmental Biology Laboratory, F-75005 Paris, France. [5] Inserm, IMRB U955-E10, F-94000 Créteil, France. [6] Université Paris Est, Faculté de médecine, Créteil, & Ecole Nationale Vétérinaire d'Alfort, F-94700 Maison Alfort, France. [7] Institute for Biochemistry, University of Cologne, D-50931 Cologne, Germany. [8] Stem Cells and Regenerative Medicine, ICAN-UMRS 1166, Université de Pierre et Marie Curie, Sorbonne Universités, F-75634 Paris, France. [9] ICAN, Institute of Cardiometabolism and Nutrition, F-75634 Paris, France. Mickael Orgeur and Sophie vom Hofe-Schneider contributed equally to this work. Correspondence and requests for materials should be addressed to S.S. (email: sigmar.stricker@fu-berlin.de)

Skeletal muscles are composed of cells of different developmental origins. In vertebrate limbs, muscle fibres as well as muscle stem cells (termed satellite cells in the adult) originate from the somitic mesoderm[1–3], while muscle connective tissue (MCT) originates from the lateral plate mesoderm[1, 2]. A mutual interdependence and cross-talk between both cell types exist during developmental and adult regenerative myogenesis. During development the myogenic progenitors migrate from the somites to the nascent limb buds, where they form initial pre-muscle masses that amplify and then subdivide into individual muscles with each muscle having an individual shape and size[4, 5]. It is widely accepted that vertebrate myogenic cells do not contain intrinsic information that govern their place and time of differentiation[6, 7]. Instead, there is long-standing evidence that limb muscle patterning is mediated by extrinsic signals from local lateral plate mesoderm derived cells[8–10]. This non-cell autonomous function is most likely mediated by MCT cells providing local but still undefined cues for myogenic cell proliferation, differentiation, survival and local migration[11–13].

MCT is often used as a collective term for different interstitial cell types, i.e. cells that can be found between the myofibres in adult muscle. Muscle interstitial cells comprise Tcf4[+] connective tissue fibroblasts that originate from lateral plate mesoderm[12, 14], but also diverse populations of mesenchymal progenitor cells have been defined[15–18]. The exact relationship between connective tissue fibroblasts and interstitial mesenchymal progenitors is unclear and it was proposed that they largely overlap[14, 19]. Amongst these progenitors, a population of non-myogenic fibro-adipogenic progenitors (FAPs) was identified that rapidly expand upon muscle injury and promote myogenesis in vitro[20], but also give rise to fibrosis and fatty infiltration in diseases[21, 22]. The pro-myogenic function ascribed to FAPs during adult muscle regeneration is an intriguing parallel to the function lateral plate-derived MCT progenitors play during embryonic myogenesis. Whether or not FAPs and developmental MCT progenitors are related and what molecular mechanisms contribute to their pro-myogenic function is mostly unknown.

In this study we show that the transcription factor Odd skipped-related 1 (Osr1) marks a subset of embryonic MCT cells that constitute a developmental FAP-like population, which supports embryonic myogenesis and is also a developmental source of adult muscle interstitial FAPs. We further show that loss of Osr1 function during limb development leads to a marked decrease in myogenic progenitor expansion and muscle patterning defects. Consistent with these observations, we show that Osr1 lies upstream of a large number of genes that control muscle connective tissue function.

## Results

**Osr1 marks a subpopulation of limb MCT cells.** *Osr1* is expressed in MCT cells in chick embryos and is involved in chick connective tissue differentiation[23, 24]. In mice, *Osr1* expression can be followed via an eGFP-IRES-CreERt2 knock-in allele ($Osr1^{GCE}$)[25] and was found in embryonic limb mesenchyme correlating with areas of myogenic differentiation between E11.5 and E13.5 (Fig. 1a–e, Supplementary Fig. 1a). Apart from the limbs, Osr1[+] interstitial cells were also observed in and surrounding other muscles as the shoulder girdle muscles and superficial as well as deep muscles of the back (Supplementary Fig. 1b). Not all muscles harbour Osr1[+] interstitial cells; low GFP signal was seen, for example, in intercostal muscles (Supplementary Fig. 1b). In addition, the diaphragm and the tongue contained Osr1[+] interstitial cells (Supplementary Fig. 1c). In the limbs, Osr1[+] cells were observed in close association with Lbx1[+] and Pax7[+] myogenic progenitors (Fig. 1a, b), closely associated

with but distinct from Myod1[+] myoblasts (Fig. 1c) and differentiating myofibres (Fig. 1d). At E14.5, Osr1[+] cells were found in an interstitial position between myofibres (Fig. 1e). The *Osr1*-GFP expression pattern was similar to that of *Osr1* transcripts (Fig. 1f). *Osr1* expression was independent of the presence of myogenic cells as shown by normal *Osr1* expression in muscleless limbs of $Pax3^{GFP/GFP}$ mutant mice (Fig. 1f, Supplementary Fig. 2).

Freshly FACS-isolated Osr1[+] cells from E13.5 $Osr1^{GCE/+}$ embryonic limbs (Fig. 1g) expressed fibroblastic markers, including type I Collagen (COL1), PDGFRα, Vimentin, type VI Collagen (COL6) and alpha-smooth muscle actin (αSMA) (Fig. 1h). This fibroblastic signature was maintained in cultures with standard growth medium (Fig. 1i). Consistent with these observations, mRNA-sequencing (RNA-Seq) data from $Osr1^{GCE/+}$ cells revealed that freshly isolated Osr1[+] cells strongly express genes encoding ECM components characteristic of fibroblasts (Supplementary Table 1). To exclude Osr1 expression in a small subset of myogenic cells or low-level expression in myogenic cells, we FACS-isolated limb mononuclear cells divided into *Osr1*-expressing (GFP[+]) and *Osr1*-negative (GFP[−]) populations (Supplementary Fig. 3a) and analysed myogenic marker expression after cytospin. Myf5[+] or Myogenin[+] cells were only found in the GFP[−] FACS fraction, anti-GFP antibody labelling did not identify any GFP[+]/Myf5[+] or GFP[+]/Myogenin[+] cells (Supplementary Fig. 3b). Altogether, this establishes that Osr1 is a marker for a subpopulation of mouse limb MCT cells.

**Osr1[+] MCT population partially overlaps with Tcf4.** Tcf4 (Tcf7l2) is a recognised marker for embryonic MCT cells[11, 12], however less than 60% of FACS-isolated Osr1[+] cells showed Tcf4 expression at E13.5 (Fig. 1h, i), indicating that limb MCT is a heterogeneous entity that contains distinct subpopulations. We then compared the expression of Osr1 and Tcf4 in E11.5 and E13.5 hindlimbs (Fig. 2a, b). At E11.5 Osr1 and Tcf4 were expressed in limb mesenchyme flanking the cartilaginous condensation of the humerus (Fig. 2a). The Osr1 or Tcf4 expression domains partially overlapped, cells expressing both transcription factors were found, but also cells expressing either Osr1 or Tcf4 exclusively (Fig. 2a). Osr1 was also expressed in mesenchyme flanking the scapula condensation, where Tcf4 was not expressed (Fig. 2a). At E13.5, Osr1 and Tcf4 showed distinct expression patterns in hindlimbs, although with areas of coexpression (Fig. 2b). Osr1 was generally found more towards the periphery, while Tcf4 was prevalent in central regions. The biceps femoris muscle exclusively harboured Osr1[+] cells, the flexor digitorum longus muscle exclusively harboured Tcf4[+] cells. Interstitial cells coexpressing Osr1 and Tcf4 were, for example, observed in the gastrocnemius and tibialis anterior muscles. The gastrocnemius interstitial cells showed higher Osr1, the tibialis anterior interstitial cells higher Tcf4 expression at this stage (Fig. 2b).

We next analysed the relationship of Osr1 and Tcf4 expression at later developmental stages. To this end we isolated foetal (E18.5) *Osr1*-expressing and *Osr1*-negative (GFP[+] and GFP[−]) interstitial cells from dissected hindlimb muscles by FACS (CD45[−];CD31[−];Ter119[−]; α7-integrin[−] to remove satellite cells) and analysed Tcf4 expression in GFP[+] and GFP[−] fractions via immunolabelling after cytospin. The majority of the GFP[+] cells were Tcf4[+] (72,9%). The GFP[−] fraction harboured apparently less Tcf4[+] cells (64,3%). The low overlap of both markers at E13.5 (Fig. 1h, Fig. 2b) suggests that Osr1[+] cells acquire Tcf4 expression during development. In addition, the majority of the GFP[+] cells were positive for PDGFRα (72,1%), while the GFP[−] pool was clearly reduced for PDGFRα[+] cells (43,7%; Fig. 2c). Of all Tcf4[+] cells (i.e. within GFP[+] and GFP[−] populations), 66,4% expressed PDGFRα (Fig. 2c). This establishes that neither Tcf4 nor Osr1 are

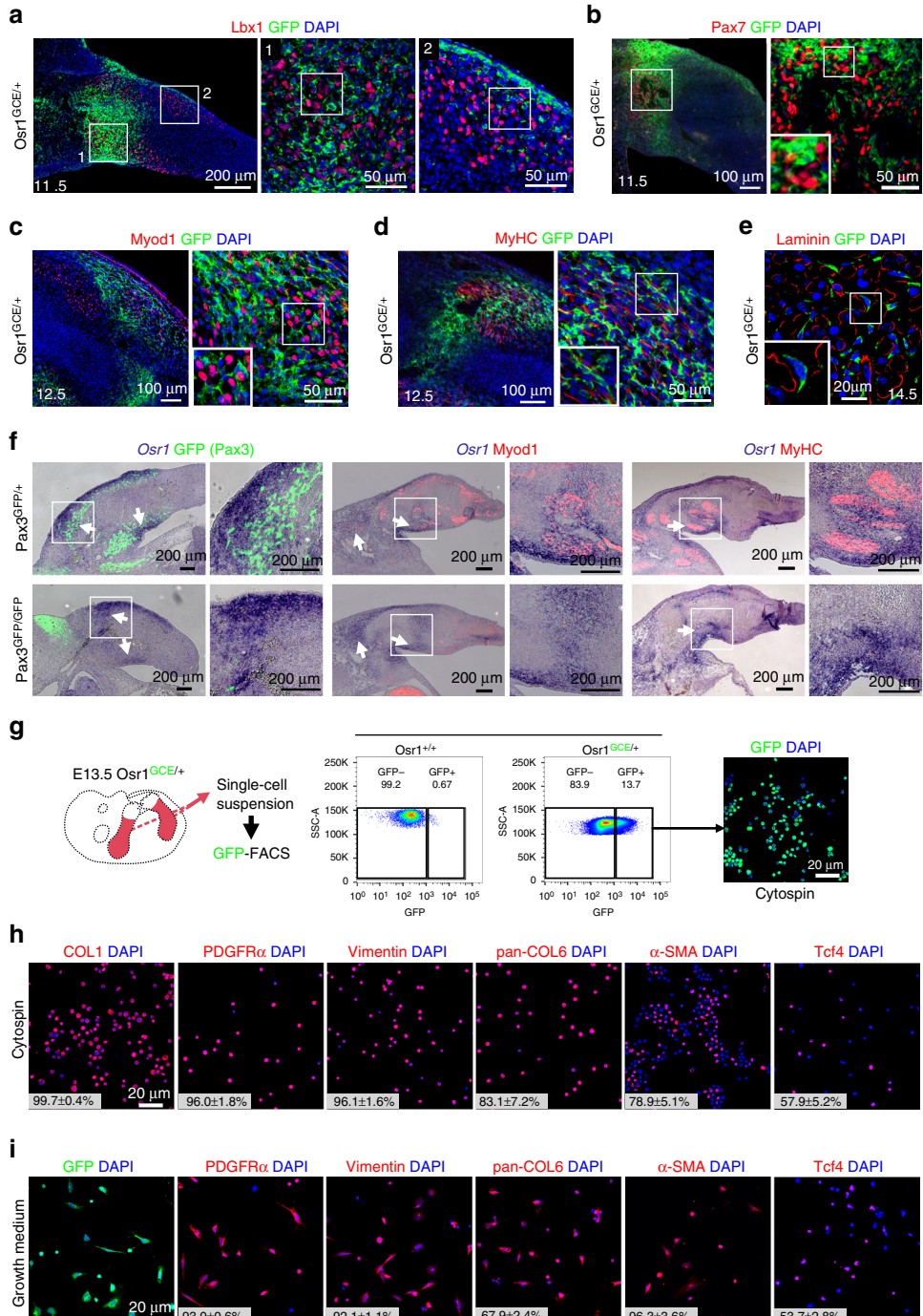

**Fig. 1** Osr1 is expressed in limb mesenchymal cells associated with myogenic cells and labels MCT cells. **a–e** *Osr1* expression was assessed by immunolabelling for GFP on *Osr1*[GCE/+] embryos at the indicated stages. Osr1[+] cells are found interstitial to Lbx1[+], Pax7[+] and MyoD1[+] myogenic progenitors and interstitial to developing myofibres labelled either for Myosin heavy chain (MyHC) or Laminin. Note that in some developing muscles Osr1[+] cells are abundant, while being scarce in others. Boxed regions (numbered in **a**) are shown in the panels to the right of overview images and are magnifications of the area. **f** *Osr1* expression was analysed by in situ hybridization to sections of muscleless limbs of E11.5 (left), 12.5 (middle) and 13.5 (right) *Pax3* mutant embryos. The overall expression pattern of *Osr1* is similar in limbs of *Pax3*[GFP/+] and *Pax3*[GFP/GFP] embryos. Arrows highlight strong *Osr1* expression observed in control and muscleless limbs. Boxed regions are shown in the panels to the right of overview images and are magnifications of the area. **g** Schematic depiction of limb Osr1[+] cell isolation procedure and examples of FACS plots. **h** Expression of fibroblastic markers in freshly FACS-isolated Osr1[+] cells subjected to cytospin. ($n = 3$). **i** Osr1[+] cells cultured in growth medium maintained the expression of fibroblastic markers ($n = 3$). Insets in **h**, **i** are percentage of COL1[+], PDGFRα[+], Vimentin[+], Pan-COL6[+], αSMA[+] and Tcf4[+] cells among the Osr1[+] cell population. Values represent mean ± s.e.m. N-numbers indicate biological replicates, i.e. samples from different specimen

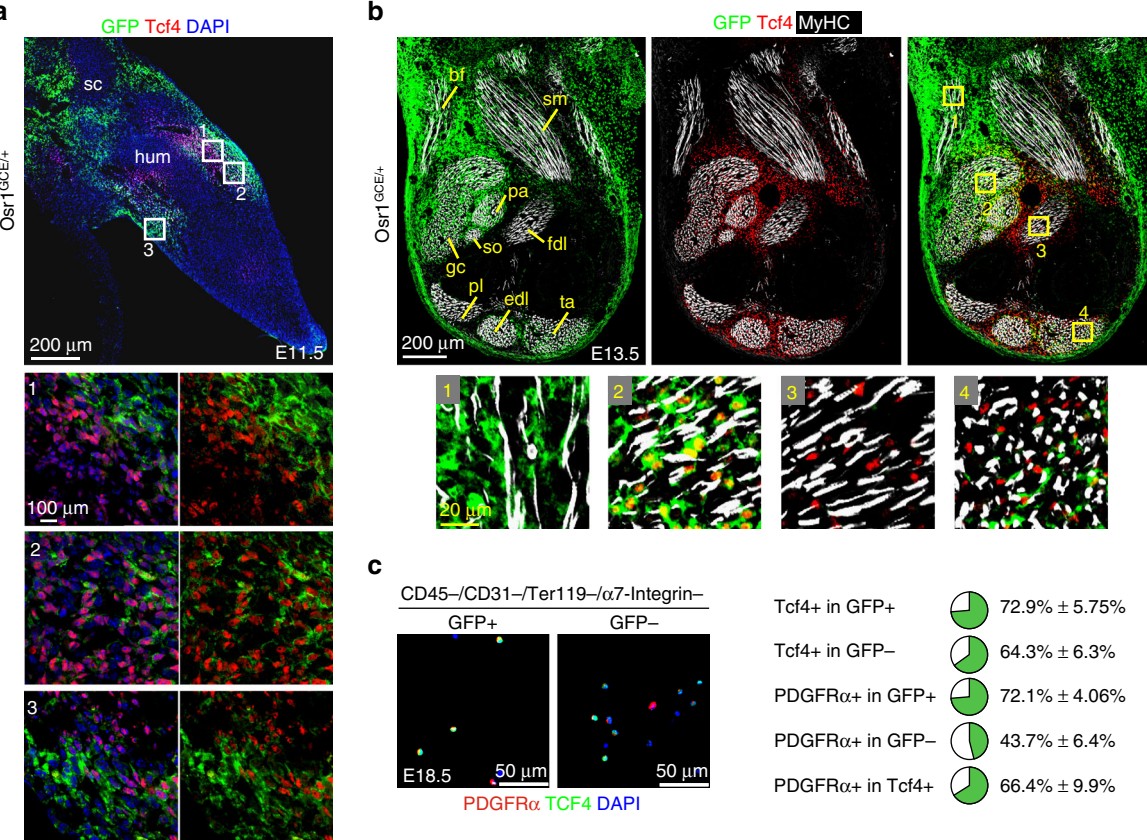

**Fig. 2** Osr1 and Tcf4 identify distinct and common MCT cells. **a, b** *Osr1* expression was assessed by immunolabelling for GFP on E11.5 and E13.5 *Osr1GCE/+* embryos. Boxed regions are shown in the numbered panels below and are magnifications of the area. **a** At E11.5 Osr1 is expressed in limb mesenchyme at the level of the humerus condensation overlapping with Tcf4. Tcf4 is expressed towards the centre and Osr1 towards the periphery of the limb. **b** At E13.5 Osr1 is expressed in limb mesenchyme in the periphery and encompassing the biceps femoris (bf), gastrocnemius (gc), soleus (so), plantaris (pa), semimembranosus (sm), peroneus longus (pl), extensor digitorum longus (edl) and tibialis anterior (ta) muscles. Tcf4 is expressed in central areas of the limb, where it is exclusive in the flexor digitorum longus (fdl) muscle, and within and surrounding the edl and ta muscles. **c** E18.5 limb muscle interstitial cells were FACS-isolated as indicated and separated into *Osr1*-expressing and *Osr1*-negative (GFP$^+$ and GFP$^-$) populations. Cytospun cells were labelled for Tcf4 and PDGFRα. Quantification showing the percentages of PDGFRα$^+$ and Tcf4$^+$ cells in GFP$^+$ or GFP$^-$ populations is shown below. Percentage of PDGFRα cells in all Tcf4 cells was counted in both GFP$^+$ and GFP$^-$ populations (*n* = 3). Values represent mean ± s.e.m (*n* = 3). N-numbers indicate biological replicates, i.e. samples from different specimen

comprehensive MCT cell markers, but indeed appear to identify region-specific subpopulations of MCT in development.

**Osr1$^+$ cells are developmental fibro-adipogenic progenitors.** Limb lateral plate derived cells are a heterogeneous population exhibiting multiple differentiation potentials. We therefore analysed the differentiation potential of the Osr1$^+$ cell population. FACS-isolated Osr1$^+$ cells exposed to fibrogenic, adipogenic, chondrogenic, osteogenic or myogenic conditions in vitro, displayed robust fibrogenic and adipogenic differentiation, whereas Osr1$^+$ cells displayed very low chondrogenic and no detectable osteogenic or myogenic potentials (Fig. 3a). To analyze Osr1$^+$ cell differentiation in vivo, we pulsed Osr1$^{GCE/+}$;R26R$^{mTmG}$ embryos with a single dose of Tamoxifen (TAM) at E11.5 or E13.5 followed by analysis after organogenesis (E18.5). This in vivo genetic Osr1$^+$ cell lineage tracing showed a contribution of Osr1$^+$ cell progeny in limb muscle interstitial regions and dermis at E18.5 (Fig. 3b, Supplementary Fig. 4a). Consistent with in vitro adipogenic differentiation (Fig. 3a), Osr1$^+$ cell progeny were observed in subcutaneous white adipose tissue in vivo (Fig. 3b, Supplementary Fig. 4a).

As shown above, specific muscles harbour different amounts of Osr1$^+$ cells (Fig. 2b). We thus analysed the contribution of E11.5

Osr1$^+$ cells to PDGFRα$^+$ and Tcf4$^+$ interstitial cells in the gastrocnemius muscle as an example for high Osr1 expression, and the tibialis anterior as an example for lower Osr1 expression. As expected, Osr1 descendants labelled by a single TAM dosage at E11.5 contributed stronger to PDGFRα$^+$ or Tcf4$^+$ interstitial cells in gastrocnemius than tibialis anterior (Fig. 3c). Taken together, our data reveal that Osr1$^+$ cells represent spatially allocated mesenchymal progenitors with fibrogenic and adipogenic fates during development resembling adult muscle interstitial FAPs[20, 21].

Adult mouse FAPs are characterized by expression of the surface markers Sca1 (Ly6A/E)[20] or PDGFRα[21, 22]. PDGFRα was also used for FAP selective isolation from human muscles[26]. In addition to these markers, adult FAPs share CD34 with satellite cells[20] and CD29, CD90 and CD166 with mesenchymal stem cells (MSCs)[21, 26]. FAPs were thus proposed to constitute a muscle-specific and lineage-restricted form of MSC[16], although this relation awaits clarification. To further elucidate the relationship of embryonic (E13.5) Osr1$^+$ cells with adult FAPs and MSCs, we analysed their surface marker profile. E13.5 Osr1$^+$ cells were mostly positive for PDGFRα (Fig. 1h, Supplementary Fig. 5a, b). Osr1$^+$ cells were positive for CD29, CD166 and mostly negative for CD90. In addition, Osr1$^+$ cells were positive for the MSC markers CD73,

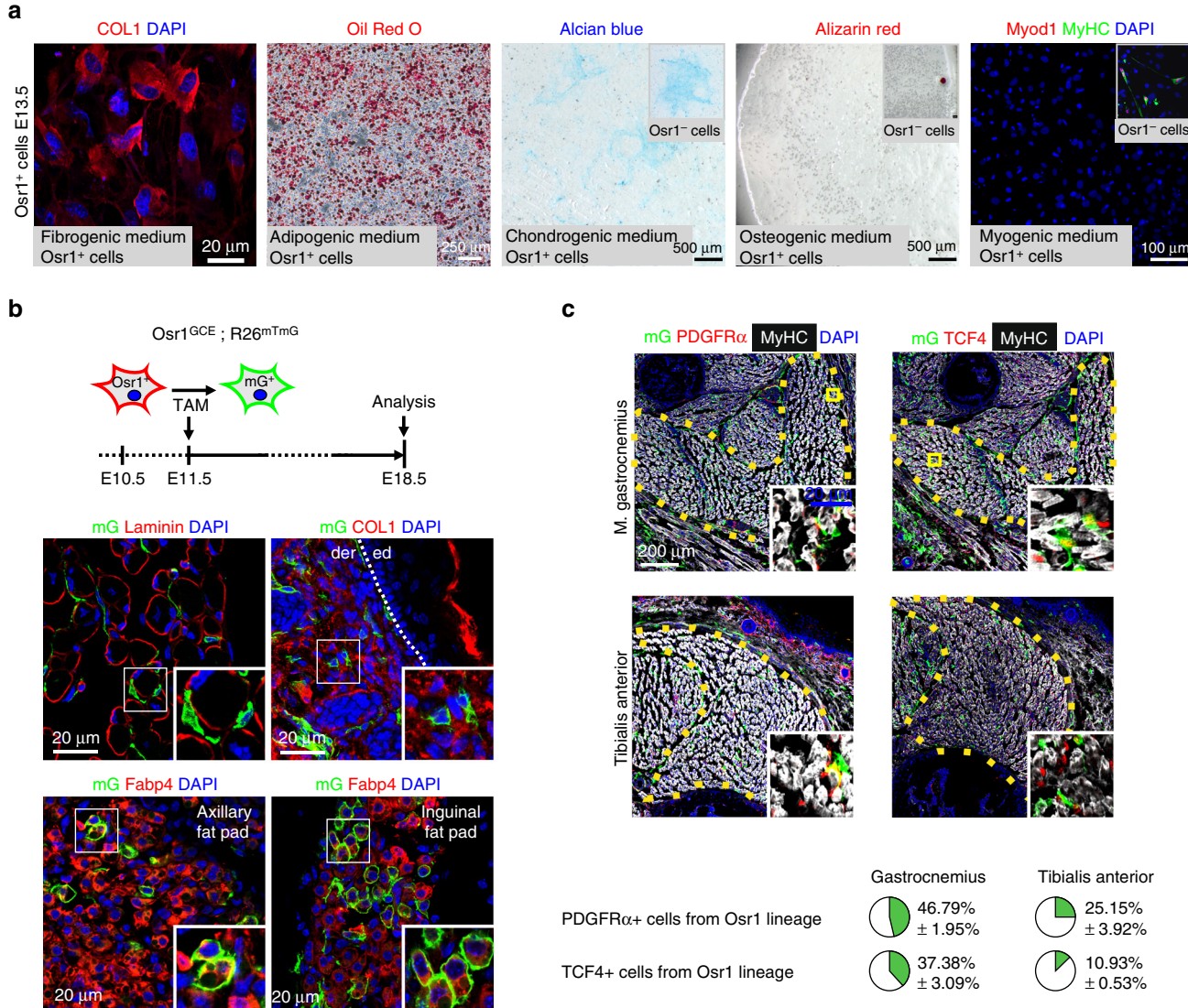

**Fig. 3** Osr1+ cells are embryonic fibro-adipogenic progenitors. **a** Cultures of Osr1+ cells under fibrogenic, adipogenic, chondrogenic, osteogenic or myogenic conditions. Fibrogenic differentiation was assessed by COL1 immunolabelling, cells exhibited stretched myofibroblast morphology. Adipogenic differentiation was assessed by Oil Red O staining, chondrogenic and osteogenic differentiation was assessed by Alcian blue and Alizarin red staining, respectively and myogenic differentiation was tested by immunolabelling for Myod1 and MyHC. Osr1+ cells show fibrogenic and adipogenic differentiation, but no osteogenic or myogenic differentiation. GFP⁻ FACS-sorted cells (Osr1⁻) containing osteo-chondrogenic and myogenic progenitors were used as controls (inserts). **b** Osr1 cell lineage tracing in E18.5 $Osr1^{GCE/+};R26R^{mTmG/+}$ embryos after Tamoxifen induction at E11.5. Membrane eGFP (mG) positive cells represent Osr1+ cell progeny. Schematic depiction of $Osr1^{GCE/+};R26R^{mTmG/+}$-based lineage tracing during development is shown above. Osr1+ progenitors give rise to muscle interstitial cells, dermis cells embedded in Collagen type I-rich matrix, and adipocytes immunolabelled for Fatty acid binding protein 4 (FABP4) in axillary and inguinal white fat pads. Ed = epidermis, der = dermis. **c** Osr1 cell lineage tracing in E18.5 $Osr1^{GCE/+};R26R^{mTmG/+}$ embryos after Tamoxifen induction at E11.5 as in **b**. Lineage contribution of Osr1 descendants (mG⁺) to PDGFRα and Tcf4 cells was analysed in the gastrocnemius and tibialis anterior muscles. Quantification is shown below (n = 3). Values represent mean ± s.e.m. N-numbers indicate biological replicates, i.e. samples from different specimen

CD105 and CD106, but negative for CD44 and CD146 (Supplementary Fig. 5a). The expression of cell surface markers was in part paralleled by their mRNA abundances in Osr1+ cells (Supplementary Table 2). Altogether the Osr1+ cell surface marker profile strongly overlaps with that of FAPs and MSCs. However, embryonal Osr1+ cells were negative for Sca1 and CD34 (Supplementary Fig. 5a). This is in line with embryonic expression of *Sca1* confined to the tail and aorta-gonad-mesonephros region[27], and apparent *CD34* expression exclusively in endothelium (Eurexpress database http://www.eurexpress.org)[28] at that stage.

We then analysed FACS-isolated foetal E18.5 muscle interstitial cells (CD45⁻;CD31⁻;Ter119⁻;α7-integrin⁻) for Sca1 expression and found that a small population of Sca1⁺ cells could be detected at this stage (Supplementary Fig. 6a). Intriguingly, Osr1⁺ cells were equally distributed between Sca1⁻ and Sca1⁺ MCT cells (Supplementary Fig. 6a). It is therefore possible that Osr1⁺ cells gain Sca1 expression over time. We then tested adult CD45⁻,CD31⁻, Ter119⁻, a7-integrin⁻, Sca1⁺ FAPs[20] for GFP expression and found very low GFP expression (close to the detection limit) in a small fraction of FAPs, but not in the other interstitial cell populations (Supplementary Fig. 6b). Analysing the expression of *Osr1*-GFP over time showed that expression of Osr1 declined in muscle interstitial cells soon after birth (Supplementary Fig. 6c). Based on surface marker profiles and in vitro and

in vivo fibro-adipogenic differentiation potentials of Osr1$^+$ cells, we conclude that Osr1$^+$ cells correspond to an embryonic FAP-like population that feeds into fibrous and adipose tissues during development. The acquisition of Sca1 expression seen in Osr1$^+$ cells at E18.5 indicate a direct continuum between Osr1$^+$ developmental cells and adult Sca1$^+$ FAPs.

**Developmental Osr1$^+$cells are a source of adult FAPs**. We next used long-term genetic lineage tracing to determine whether adult FAPs are the lineage progeny of developmental Osr1$^+$ cells. Shortly after birth, interstitial PDGFRα$^+$ cells still abundantly express Osr1 in muscles like the gastrocnemius (Fig. 4a). Genetic labelling of Osr1$^+$ cells at birth (TAM at p0 and p1) resulted in

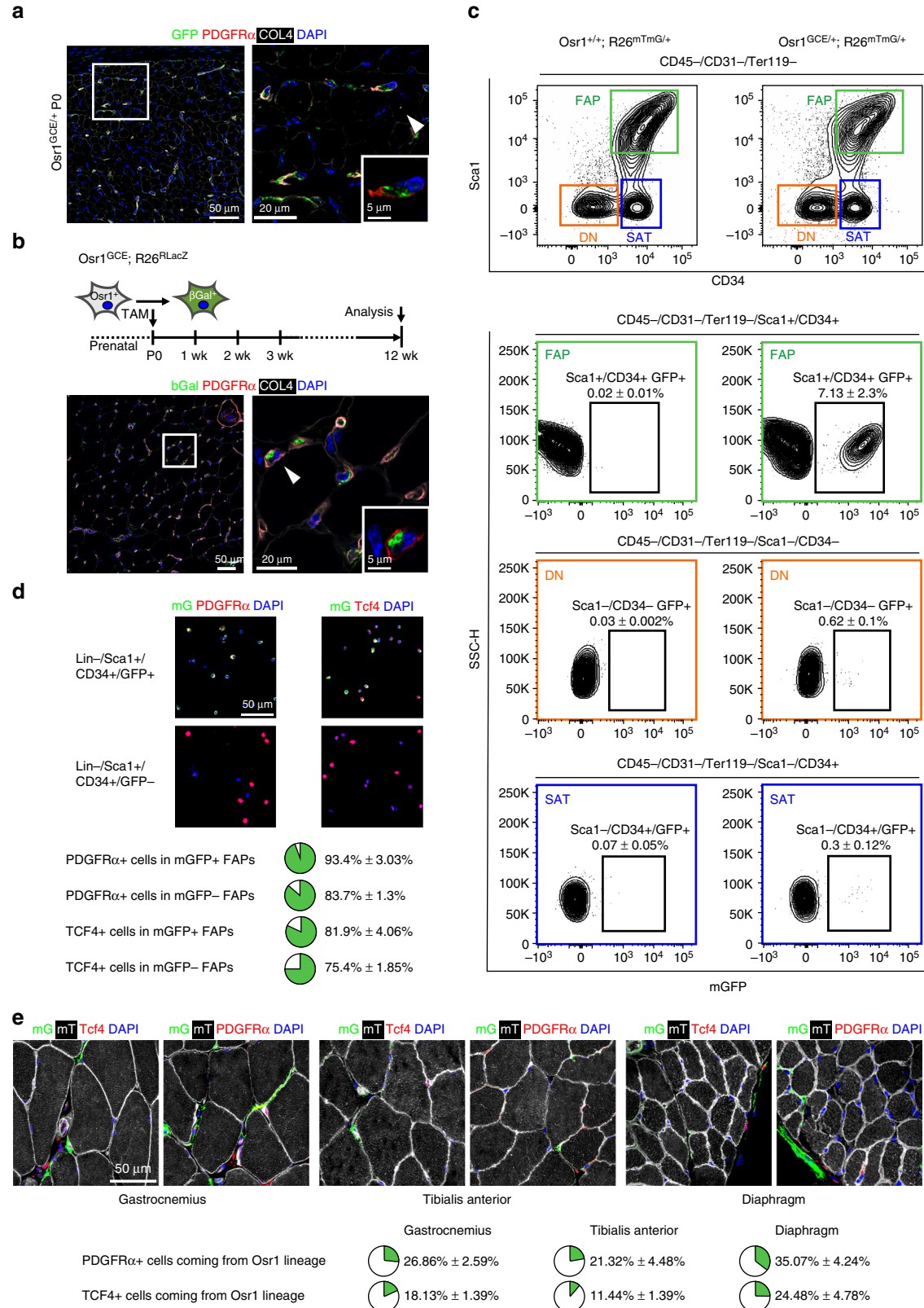

positive labelling for muscle interstitial PDGFRα+ cells in the adult (Fig. 4b). Next, we FACS isolated adult FAPs according to Joe et al.[20] from whole hindlimb muscles of Osr1GCE/+;R26mTmG or, as negative control, Osr1+/+;R26mTmG animals Tamoxifen-induced at p0/p1. FAPs (CD45−;CD31−;Ter119−;Sca1+;CD34+), satellite cells (CD45−;CD31−;Ter119−;Sca1−;CD34+) and osteo-chondrogenic cells (CD45−;CD31−;Ter119−;Sca1−;CD34−) were analysed for mGFP positive cells. Osr1 progeny was observed exclusively in FAPs, while no lineage progeny was found in satellite cells or osteo-chodrogenic progenitors (Fig. 4c) showing that Osr1+ progenitors only contribute to the FAP pool. Analysis of cytospun mGFP+ and mGFP− FAPs showed that almost all mGFP+ cells are PDGFRα+. In addition, mGFP+ and mGFP− FAPs harbour a high proportion of Tcf4+ cells (Fig. 4d). The mGFP+ population interestingly appeared enriched for both markers (Fig. 4d). To confirm these findings, the progeny of Osr1+ cells was analysed on sections of gastrocnemius and tibialis anterior muscles and, as an example for a non-limb muscle, the diaphragm. 21% (TA), 26% (gastrocnemius) and 35% (diaphragm) of interstitial PDGFRα+ cells originated from developmental Osr1+ cells pulsed at p0/p1 (Fig. 4e). In addition, Osr1+ progenitors also contributed to different extent to Tcf4+ cells (Fig. 4e). This correlates with the embryonic lineage tracing results obtained for the TA and gastrocnemius muscles at E18.5 (Fig. 3c) and suggests that the level of Osr1+ cell contribution to adult FAPs in individual muscles is reflected by the developmental Osr1 expression pattern. Altogether, this establishes that adult FAPs derive, at least in part, from developmental Osr1+ cells.

**Osr1 regulates embryonic myogenesis.** Given the supportive function of adult FAPs in regenerative myogenesis and the non-cell autonomous function ascribed to limb MCT cells during muscle development, we asked whether Osr1+ FAP-like cells have a similar supportive function during embryonic myogenesis. Homozygous Osr1GCE/GCE mutants die between E14.5 and E15.5 due to heart defects[29], which allowed us to analyse embryonic myogenesis. 3D reconstruction of limb muscles at E14.5 revealed specific muscle patterning defects in Osr1 mutants. In hindlimbs, stylopod muscles were truncated (Fig. 5a). The biceps femoris accessory and posterior head muscles were shortened and myo-fibres were misoriented compared to wildtype (Fig. 5a). Sections showed a complete absence of myogenic cells in distal areas of the biceps femoris (Fig. 5b), suggesting that muscle progenitors either failed to reach the distal positions or did not survive. In contrast to muscle truncation, scattered ectopic myogenic cells and muscle fibres were observed between muscles (Fig. 5c, Supplementary Fig. 7c, d). Several other muscles of hind- and forelimbs as well as the shoulder and pelvic girdle were variably affected (Supplementary Fig. 7a–e). Overall muscles showing high Osr1 expression such as the biceps femoris (Fig. 5a) or gastrocnemius muscles (Supplementary Fig. 7e) showed clear aberrations, while muscles with low Osr1 expression as the tibialis anterior, extensor digitorum longus or peroneus longus muscles appeared mostly unaffected (Supplementary Fig. 7d).

Myod1 and Myog (Myogenin) expression in E13.5 forelimbs showed defects in size, shape and/or trajectory of individual muscle anlagen in Osr1 mutants (Fig. 5d, Supplementary Fig. 7f). In addition, nascent muscle mass anlagen were less sharply defined as compared to wildtype mice (Fig. 5d). This was already noticeable at earlier stages of development (E11.5), where myogenic cells were present in regions that normally do not contain myogenic cells, and myogenic anlagen displayed a reduced myogenic cell density per area in Osr1 mutants (Supplementary Fig. 8a). This indicates a partial misdistribution of myogenic cells in Osr1 mutants in line with the local lack of myoblasts or local appearance of ectopic myoblasts shown above. Maintenance and amplification of the initial myogenic progenitor pool is essential for correct muscle patterning. In Osr1 mutant limbs, the myogenic progenitor pool was reduced at E11.5 and E13.5 compared to respective wildtypes (Fig. 5e, Supplementary Fig. 8b, e). The number of myogenic progenitors was equally reduced in the proximal as well as dorsal and ventral limb muscle masses at E11.5 indicating that the initial migration step of muscle progenitors was not overtly impaired (Fig. 5e). A measurement of the ratios of Pax7+ vs. Myod1+ cells or Pax7+/Myod1+ cells vs total Pax7+ cells showed no difference during early myogenic differentiation in Osr1 mutants at E11.5 (Supplementary Fig. 8b) indicating that the reduction in the number of myogenic cells was not caused by precocious differentiation. Myogenic cells (Lbx1+, Pax7+, Myod1+) of E11.5 (Fig. 5f, Supplementary Fig. 8c, d) and E13.5 (Supplementary Fig. 8f) Osr1 mutants showed a decreased proliferation rate. In addition, myogenic cells showed a significantly increased apoptotic rate in E11.5 Osr1 mutants as compared to wildtype (Fig. 5g). We conclude that a loss of Osr1 function in embryonic FAPs leads to limb muscle patterning defects. These defects are preceded by a reduction of the muscle progenitor pool caused by decreased proliferation and survival of myogenic cells in combination with a misdistribution of myogenic cells. The limb muscle phenotype was not caused by an absence of Osr1+ cells (Supplementary Fig. 7g, h).

**Osr1 maintains muscle connective tissue identity.** To identify Osr1 target genes underlying the non-cell autonomous effect of Osr1+ cells on muscle formation, we analysed the transcriptome of E13.5 Osr1GCE/GCE cells vs. Osr1GCE/+ cells. We identified 511 differentially expressed (DE) genes (Fig. 6a). Gene ontology (GO) analysis for "cellular component" highlighted enrichments for "extracellular region" and "extracellular matrix" related genes (Fig. 6b). Consistently, KEGG analysis of all DE genes yielded enrichment for "ECM-receptor interaction" (Supplementary Fig. 9a). Interestingly, genes belonging to the ECM cluster showed a bipartite behaviour with approximately half of the genes being down-regulated or up-regulated (Fig. 6c). Analyses of gene expression using the Eurexpress database[28] revealed predominant expression of ECM genes that were upregulated in Osr1 deficient cells in cartilage and/or tendons. In accordance, GO analysis of upregulated genes revealed a significant enrichment for "skeletal

---

**Fig. 4** Developmental Osr1+ cells are a source for adult FAPs. **a** Osr1 is expressed in PDGFRα+ interstitial mesenchymal progenitors at birth (postnatal day P0). **b** Osr1 cell lineage tracing in adult (P84/12 week) Osr1GCE/+;R26RLacZ/+ animals after Tamoxifen induction at P0. Osr1+ cell progeny is marked by immunolabelling for beta-Galactosidase (bGal). Perinatal Osr1+ cells give rise to interstitial PDGFRα+ FAPs. **c** Osr1 cell lineage tracing in adult (12 week) Osr1GCE/+;R26mTmG/+ animals after Tamoxifen induction at P0; Osr1+/+;R26mTmG/+ animals used as controls are shown left. FAPs (Sca1+, CD34+), satellite cells (SAT) and double-negaitve cells (DN) were FACS-isolated and analysed for mGFP expression. Only FAPs contained a clearly detectable mGFP+ population (n = 3). **d** mGFP+ and mGFP− FAPs isolated as in **c** were cytospun and analysed for PDGFRα and Tcf4 expression. Quantification is shown below (n = 3). **e** The contribution of p0 Osr1+ cells to muscle interstitial PDGFRα+ and Tcf4+ cells was quantified on sections of Osr1GCE/+;R26mTmG/+ animals in the indicated muscles. Quantification is shown below (n = 4). Values represent mean ± s.e.m. N-numbers indicate biological replicates, i.e. samples from different specimen

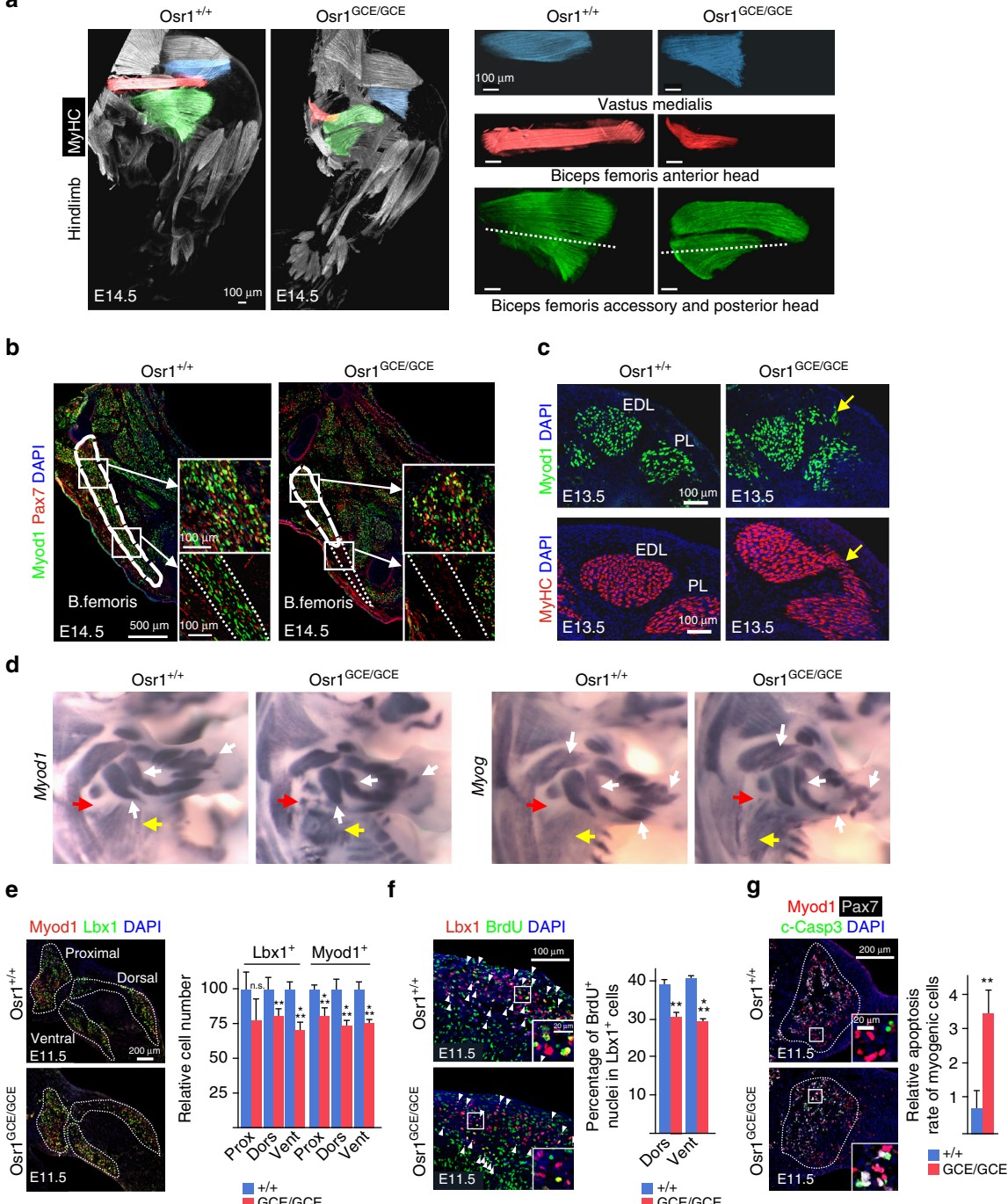

**Fig. 5** Limb muscle patterning defects in *Osr1*-deficient embryos. **a** Muscle pattern in hindlimbs of E14.5 *Osr1*[+/+] and *Osr1*[GCE/GCE] embryos assessed by whole limb immunolabelling for Myosin heavy chain (MyHC) followed by 3D reconstruction. Affected muscles are highlighted by false colours. Note truncations of several muscles. Scale bars: 100 μm. **b** Sections of E13.5 hindlimbs, where the Biceps femoris muscle is highlighted. Section planes are indicated in green muscles (a, bottom right) by the dashed lines. The distal truncated region of the B. femoris muscle is devoid of myogenic cells in *Osr1* mutants highlighted with dashed lines. In the residual B. femoris muscle of *Osr1* mutants, myogenic cells appear in disarray (see high magnifications). Analysis was done on serial sections of whole limbs to exclude section artefacts. **c** Ectopic Myod1[+] myoblasts and MyHC[+] myofibers between the Extensor digitorum longus (EDL) and Peroneus longus (PL) muscles in *Osr1* mutants (yellow arrows). **d** Wholemount in situ hybridization on E13.5 *Osr1*[+/+] or *Osr1*[GCE/GCE] embryos for myogenic markers *Myod1* and *Myog*. Misshaped muscle primordia in limbs (white arrows), disarrayed myofibres in the latissimus dorsi muscle (yellow arrows) and ectopic myoblasts (red arrows) are observed in *Osr1* mutants. **e** Immunolabelling for Lbx1 and Myod1 on E11.5 limb sections shows a decrease in the overall number of myogenic cells in proximal, dorsal and ventral muscle primordia in mutant compared to wt embryos (*n* = 3). **f** Reduced proliferation rate of Lbx1[+] myogenic progenitors (dorsal: wt 38,6±2,3%, mut 30±1,8%; ventral: wt 41,1±1,3%, mut 29,6±1,0%) in E11.5 *Osr1* mutants assessed by BrdU labelling (*n* = 4). (**g**) Increased apoptosis in myogenic cells in *Osr1* mutants assessed by immunolabelling for cleaved (activated) Caspase 3 (c-Casp3) (*n* = 3). Error bars represent s.e.m. T-test: *=*p* < 0.05; **=*p* < 0.01; ***=*p* < 0.001. N-numbers indicate biological replicates, i.e. samples from different specimen

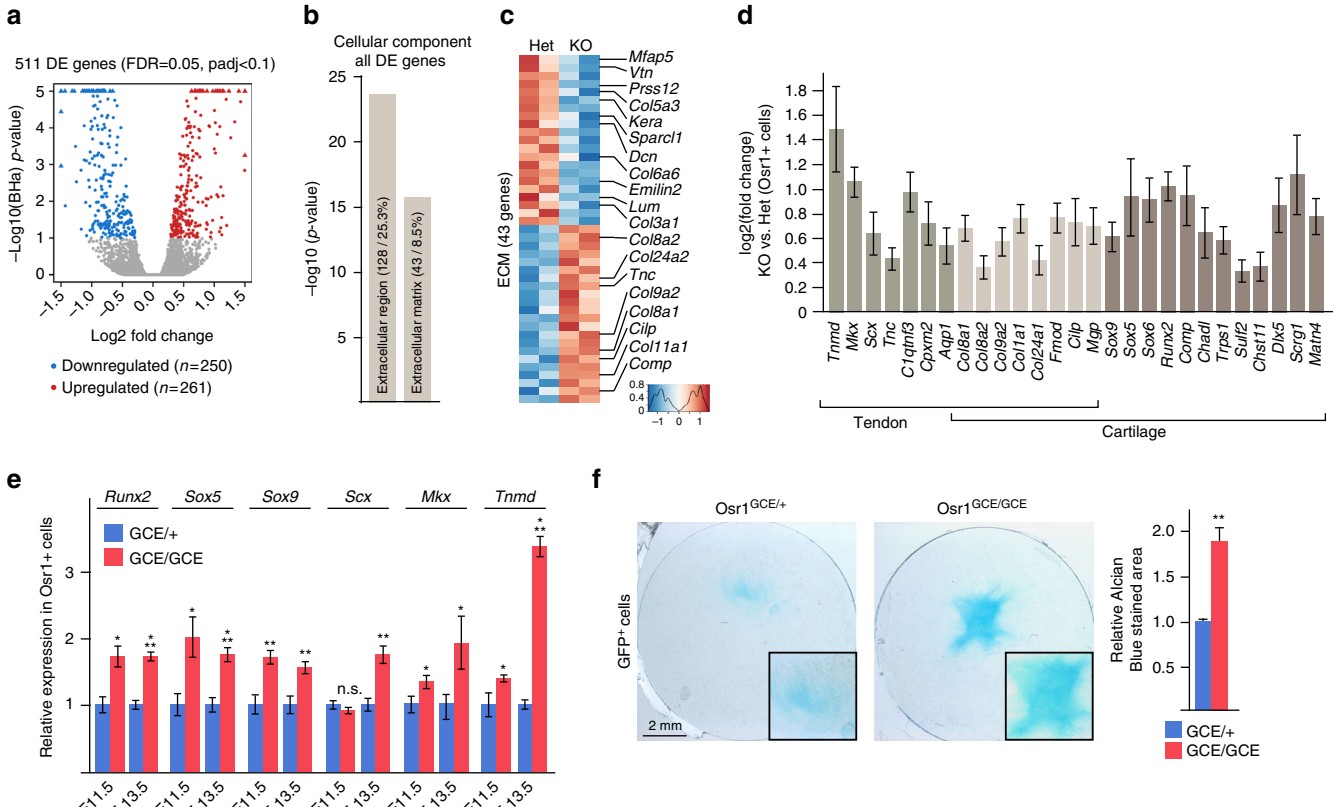

**Fig. 6** Osr1 controls ECM gene expression in a bimodal way and is required to prevent ectopic cartilage / tendon gene expression. **a** Volcano plot depiction of transcriptome analysis of FACS-sorted Osr1$^+$ cells from limbs of E13.5 Osr1$^{GCE/+}$ or Osr1$^{GCE/GCE}$ embryos shows differentially expressed (DE) genes between Osr1$^{GCE/+}$ and Osr1$^{GCE/GCE}$ limb cells. 511 DE genes were identified with a false discovery rate of 0,05 that showed a log2 fold change over 1,2 or below 0,8 and a Benjamini-Hochberg adjusted $p$-value (padj) < 0,1. **b** GO analysis of all DE genes for "cellular component" showed enrichment for the indicated terms. **c** Heatmap depiction of all genes belonging to the term "extracellular matrix" (ECM). Selected ECM genes are indicated. **d** Bar graph plot for genes selected from all DE genes for their cartilage and/or tendon- specific expression pattern in embryo; note that all cartilage/tendon genes are upregulated in Osr1 mutants. **e** Confirmation of upregulation of Runx2, Sox5 and Sox9 (cartilage genes) and Scx, Mkx and Tnmd (tendon genes) in E11.5 and E13.5 Osr1 deficient cells by RT-qPCR ($n = 4$). **f** Chondrogenic culture of Osr1$^{GCE/+}$ and Osr1$^{GCE/GCE}$ cells. Chondrogenic matrix is stained with Alcian blue, histomorphometric quantification is shown on the right. ($n = 3$). Error bars represent log2 fold change standard error (lfcSE) estimated by DESeq2 in **d** and s.e.m. in **e**, **f**. T-test: *$=p < 0.05$; **$=p < 0.01$; ***$=p < 0.001$. N-numbers indicate biological replicates, i.e. samples from different specimen

development" and "cartilage development" (Supplementary Fig. 9b). Therefore, we systematically mined the DE genes for genes connected to the GO term "cartilage development" and for tendon genes based on the list identified by Havis et al.[30]. All identified tendon and cartilage genes were upregulated in E13.5 Osr1-KO cells compared to Osr1 heterozygous cells (Fig. 6d), which was confirmed by RT-qPCR for selected genes (tendon: Scx, Tnmd, Mkx; osteo-chondrogenic: Runx2, Sox5, Sox9) (Fig. 6e). This gene deregulation was, with the exception of Scx, already detectable at E11.5 (Fig. 6e). Accordingly, Osr1 deficient cells (Osr1$^{GCE/GCE}$) showed a higher chondrogenic capacity than Osr1$^{GCE/+}$ cells in vitro (Fig. 6f). These data are consistent with the presence of ectopic cartilage differentiation in synovial joints in Osr1 and Osr2 double mutants[31], a requirement for Osr1 in preventing Sox9 expression and ectopic cartilage formation in the neural crest[32] and the ability of Osr1 to repress tendon and cartilage differentiation in chick limb mesenchymal cells[23]. We note that we also observed signs of ectopic tendon differentiation in Osr1 mutants specifically between the myo-tendinous junctions of the Teres major and Triceps brachii lateralis muscles (Supplementary Fig. 10a). Overall, Scx showed apparently increased and less confined expression in E13.5 Osr1$^{GCE/GCE}$ limbs compared to control limbs (Supplementary Fig. 10b).

In contrast to the upregulated cartilage/tendon ECM genes, ECM genes that were downregulated in Osr1 deficient cells were expressed predominantly in irregular CT and/or MCT (based on Eurexpress data). These ECM genes encoded e.g. for collagens associated to MCT, such as COL6 (Fig. 6c, Fig. 7a). COL6, typically consisting of COL6A1, COL6A2 and COL6A3 chains, is an indispensable component of the muscular ECM mutated in human muscular dystrophies[33]. Furthermore COL6 has a central role in ECM assembly, since it bridges fibrillary collagens, matrix components and cell adhesion molecules[33]. Other CT ECM components were downregulated in Osr1 mutant cells (Fig. 6c, Fig. 7a), including members of the small leucine-rich proteoglycan (SLRP) family such as Lumican (Lum) and Decorin (Dec) that play a role in ECM assembly and cell-matrix interaction[34], or the basement membrane component Nidogen 2 (Nid2). The downregulation of selected ECM genes (Col3a1, Col5a3, Col6a1, Col6a2, Col6a3, Dcn, Lum, Nid2) was confirmed by RT-qPCR (Fig. 7b). Immunolabelling for pan-COL6 revealed a strong reduction of COL6 abundance in E13.5 Osr1$^{GCE/GCE}$ limbs as compared to wildtype (Fig. 7c).

We then tested which ECM genes may be direct targets of Osr1 via Chromatin immunoprecipitation (ChIP). For this purpose freshly isolated MEFs from E13.5 C57Bl6 limbs were stably transduced with lentiviral particles expressing mouse Osr1 fused

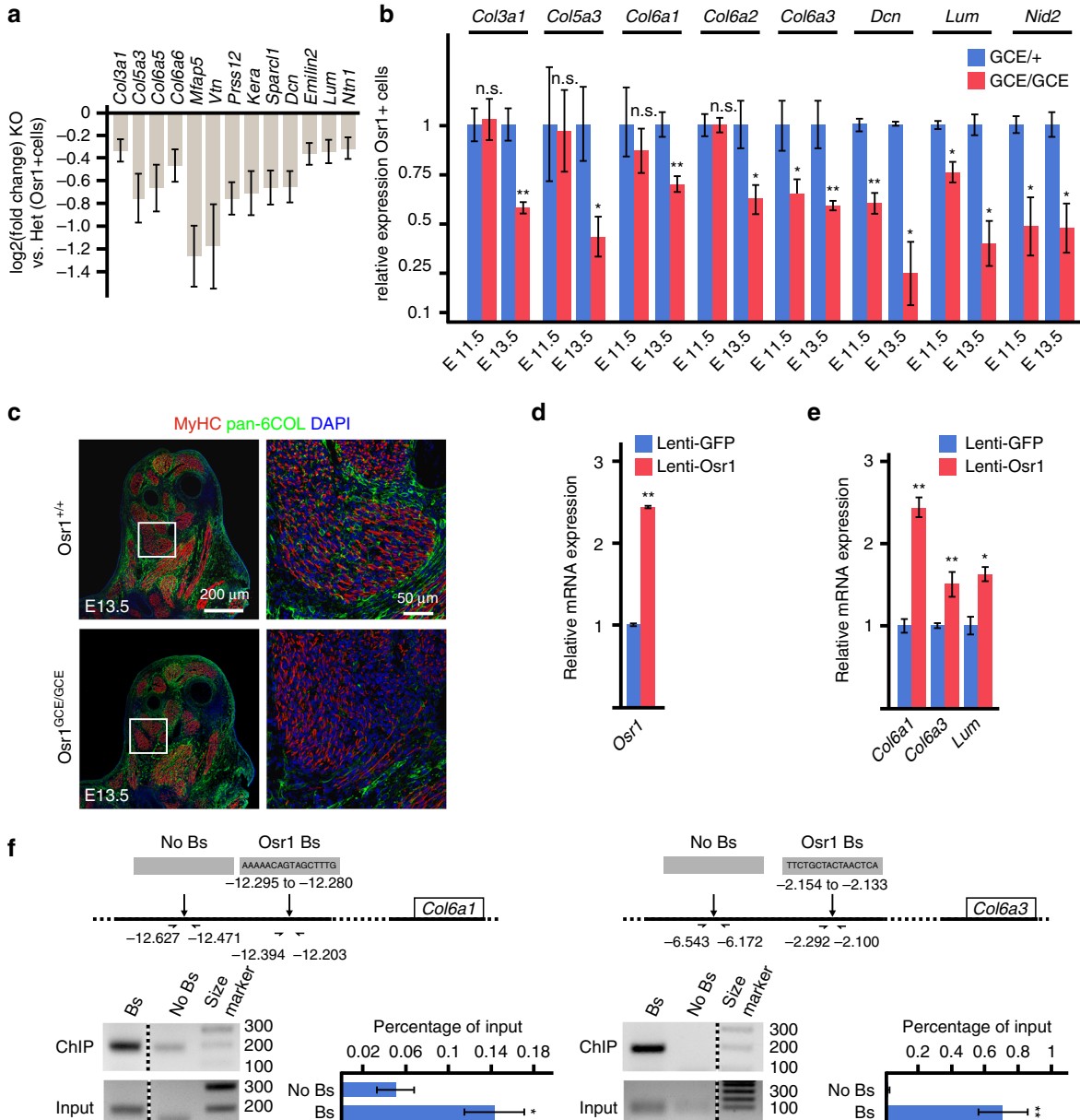

**Fig. 7** Osr1 is required for muscle connective tissue ECM gene expression. **a** Bar graph plot of genes selected from all DE genes by their expression associated with muscle connective tissue; most of the genes belong to the downregulated cluster in (Fig. 6c). **b** Confirmation of downregulation of selected MCT-ECM genes in E11.5 and E13.5 Osr1 deficient cells by RT-qPCR (n = 4). **c** Immunolabelling for MyHC and COL6 shows reduced abundance of COL6 in Osr1 mutants. **d** Transduction of E13.5 MEFs with lentiviral particles causes mild overexpression of Osr1. **e** Lentiviral overexpression of Osr1 causes upregulation of Col6a1, Col6a3 and Lum genes. **f** Chromatin immunoprecipitation of putative Osr1 binding sites at the Col6a1 and Col6a3 loci. Above: schematic representation of the loci; positions of Osr1 binding sites and PCR primers relative to exon 1 of the Col6a1 and Col6a3 genes are indicated. Below: representative gel pictures of ChIP vs. input are shown left, confirmation of binding site pulldown via RT-qPCR (n = 3) are shown right. Bs: putative binding site, No Bs: control region. Size marker: 100 base pair ladder; band sizes are indicated in base pairs. Error bars represent log2 fold change standard error (ifcSE) estimated by DESeq2 in **a** and s.e.m. in (**b**, **d**, **e**, **f**). T-test: *=p < 0.05; **=p < 0.01; ***=p < 0.001. N-numbers indicate biological replicates, i.e. samples from different specimen in **b** or independent assays in **d**, **e**, **f**.

at the 3′ end to a tripleFLAG tag under control of the low-active Ubiquitin C (UbC) promoter. This yielded mild overexpression (approximately twofold) of Osr1-tripleFLAG mRNA (Fig. 7d), which was chosen to avoid overexpression artefacts. Mild overexpression of Osr1 in MEFs was sufficient to increase the expression of Col6a1, Col6a3 and Lum mRNAs (Fig. 7e). For ChIP, upstream regions of Col6a1, Col6a3 and Lum were screened for the presence of Osr1 consensus binding sites[35, 36] in the UCSC Genome Browser[37]. Two putative binding sites were found upstream of Col6a1, Col6a1 or Lum, respectively. ChIP

yielded significant enrichment for one binding site at each the Col6a1 and the Col6a3 loci in comparison to flanking control regions without consensus binding sites (Fig. 7e, f; primer sequences for putative binding sites and controls are provided in Supplementary Table 7). These data support a direct regulation of Col6a1 and Col6a3 by Osr1. Taken together, our data support a role for Osr1 in promoting the expression of MCT-specific ECM components, while inhibiting tendon and cartilage ECM and transcription factor genes. Given that Osr1+ cells display high fibrogenic and adipogenic potential, we conclude that Osr1 is a

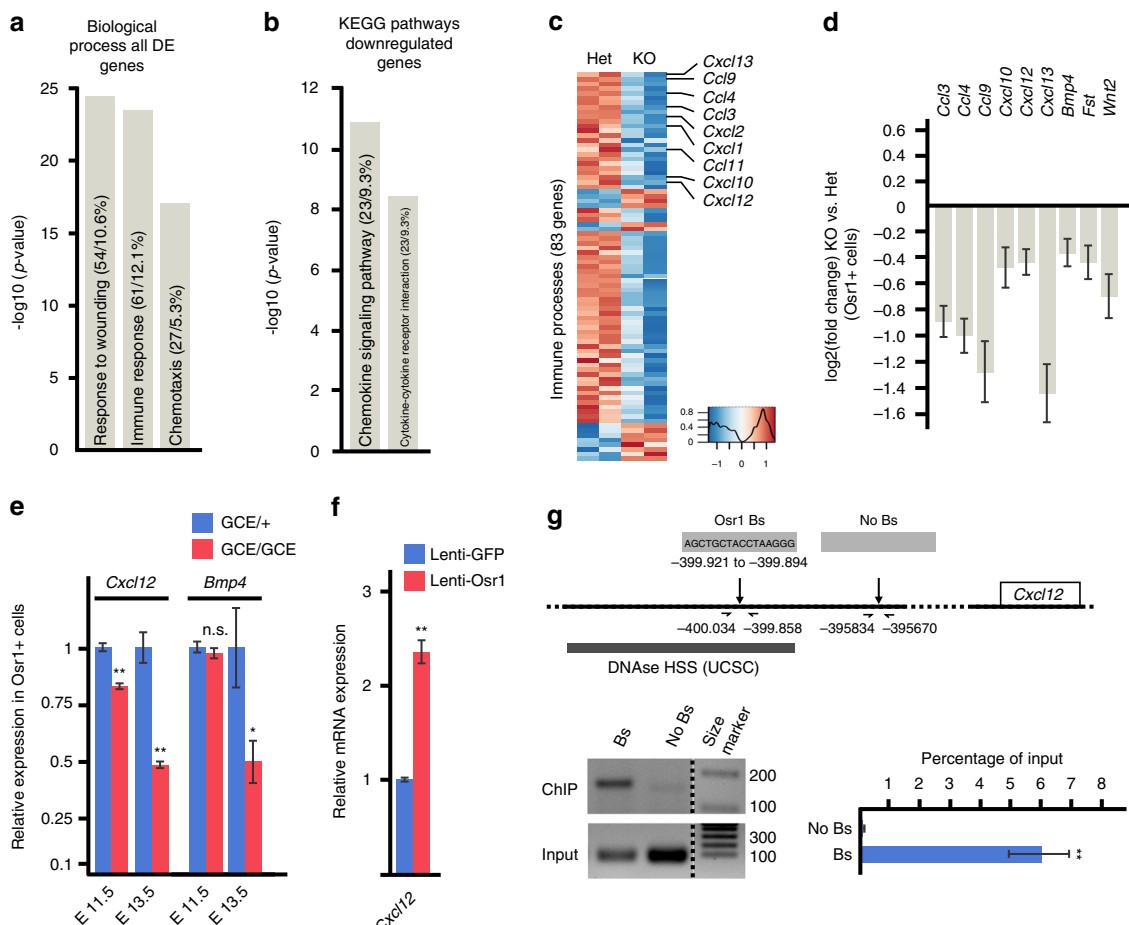

**Fig. 8** Osr1 cells are a source or growth factor signalling and Osr1 directly controls *Cxcl12* expression. **a** Gene ontology (GO) analysis for "biological process" using all deregulated (DE) genes (*Osr1^GCE/+^* vs. *Osr1^GCE/GCE^* cells) showed significant enrichment for the indicated terms. **b** KEGG pathway analysis of downregulated genes. **c** Heatmap depiction of all non-redundant DE genes contained in the GO terms shown in **c** collectively termed "immune processes". Several genes for signalling molecules from chemokine families appear downregulated in *Osr1^GCE/GCE^* limb cells. **d** Deregulation of secreted signalling molecules selected from all DE genes for their previously demonstrated role in myogenesis. **e** Confirmation of downregulation of *Cxcl12* and *Bmp4* in Osr1-deficient cells isolated at E11.5 and E13.5 by RT-qPCR (n = 5). **f** Upregulation of Cxcl12 upon lentiviral Osr1 overexpression. **g** Chromatin immunoprecipitation (ChIP) for a putative *Cxcl12* enhancer. Representative gel pictures of ChIP vs. input are shown below schematic representation of the locus, confirmation of binding site pulldown via RT-qPCR (n = 3) shown right. Bs: putative binding site, No Bs: control region. DNAse HSS: DNAseI hypersensitive regions according to UCSC. Size marker: 100 base pair ladder; band sizes are indicated in base pairs. Error bars represent log2 fold change standard error (ifcSE) estimated by DESeq2 in **d** and s.e.m. in **e**, **f**, **g**. T-test: *=$p < 0.05$; **=$p < 0.01$. N-numbers indicate biological replicates, i.e. samples from different specimen in **e** or independent assays in **f**, **g**

key transcription factor that maintains embryonic Osr1+ FAP cell identity. Loss of Osr1 in turn leads to a shift in embryonic FAP transcriptional activity from an ECM characteristic for MCT to cartilage/tendon ECM.

**Osr1+ cells are a source of secreted signalling molecules**. In addition to MCT-ECM components, we searched for DE genes encoding secreted signalling molecules in *Osr1*-deficient cells. GO analysis of all DE genes or only downregulated genes for "biological processes" showed similar enrichment for the terms "response to wounding", "immune response" and "chemotaxis" (Fig. 8a, Supplementary Fig. 9c). Consistently, KEGG analysis of all DE genes or only downregulated genes showed enrichment for "chemokine signalling" and associated pathways (Fig. 8b, Supplementary Fig. 9a). Consistently, genes associated with

these terms, containing many members of chemokine family, were mostly downregulated in *Osr1* mutants (Fig. 8c), as were genes encoding several other myogenesis-related signalling factors such as Bmp4 (Fig. 8d). The decrease in expression levels of *Cxcl12* and *Bmp4* mRNAs was confirmed by RT-qPCR in Osr1+ cells (Fig. 8e). Overexpression of Osr1 induced an increase in *Cxcl12* mRNA in MEFs (Fig. 8f). The *Cxcl12* locus showed one consensus Osr1 binding site approx. 18 kb upstream and two distant upstream consensus binding sites within DNAseI hypersensitive regions (UCSC) in the gene desert 5′ of Cxcl12. A distant 5′ element located approx. 400 kb upstream of *Cxcl12* was bound by Osr1 (Fig. 8g; primer sequences are provided in Supplementary Table 7). Together this indicated direct regulation of *Cxcl12* by Osr1 via binding to this putative enhancer element.

**Osr1⁺ FAPs provide a myogenic niche in developing limbs.** The transcriptional aberrations in *Osr1* deficient cells were in part observed as soon as E11.5 concomitant with the reduced myogenic cell pool. A significant downregulation at E11.5 was found for the MCT-ECM genes *Col6a3*, *Lum*, *Dcn* and *Nid2*. In contrast, *Col3a1*, *Col5a3*, *Col6a1* and *Col6a2* showed no significant

difference in Osr1⁺ cells isolated at E11.5 (Fig. 6b). However, COL6 expression was severely decreased in E11.5 $Osr1^{GCE/GCE}$ limb muscles (Fig. 9a) indicating that the decrease of *Col6a3* mRNAs was sufficient to affect COL6 expression in line with previous in vitro and in vivo observations[38, 39]. The COL6 decrease was accompanied by a decrease of Fibronectin (FN)

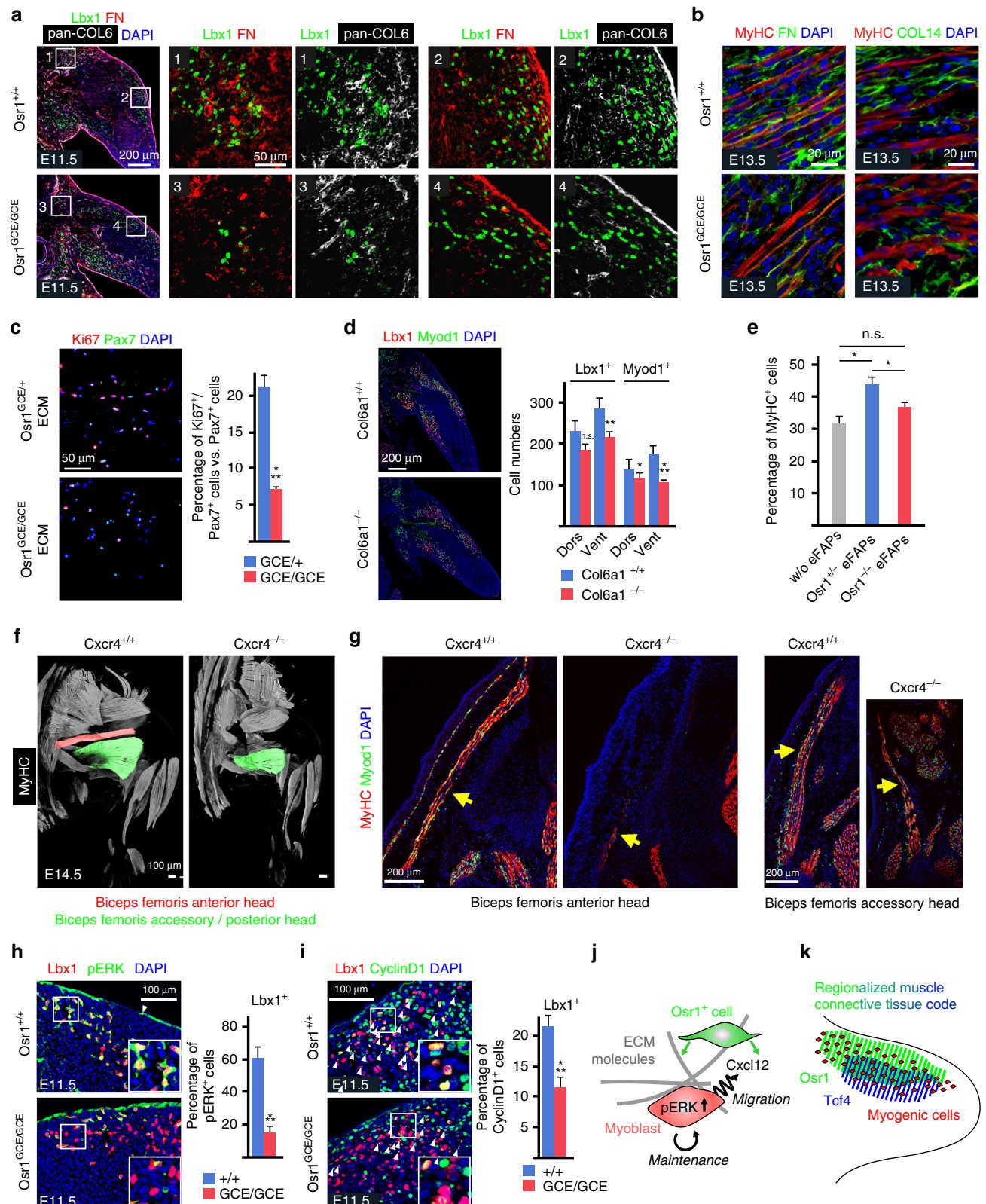

abundance (Fig. 9a), while the *Fn1* gene was not deregulated at the mRNA level (log2 fold change = 0,09). This is consistent with the impairment of FN deposition and organization in cultured fibroblasts with reduced or absent COL6 expression[40]. The reduction of FN abundance was maintained at E13.5 (Fig. 9b). Labelling for type XIV Collagen (COL14) also revealed a disruption of ECM structure (Fig. 9b). COL14 is a fibril-associated collagen with interrupted triple helices (FACIT) that is important for assembly of the fibrillar collagens and for crosslinking these to other ECM components like SLRPs such as Decorin[41]. This altogether confirms that the MCT-ECM is disorganized in *Osr1* mutants. The ECM components FN and Vitronectin (Vtn), the latter of which was downregulated in RNA-Seq (Figs. 6c, 7a), are known to enhance proliferation of cultured myoblasts[42, 43]. This emphasizes that a specific composition of the MCT-ECM is crucial for regulating embryonic myoblast proliferation. To analyse if Osr1 cells could influence myoblast proliferation via their ECM production, we cultured FACS-isolated Osr1$^{GCE/+}$ and Osr1$^{GCE/GCE}$ cells for 10 days to produce an ECM substrate on the culture plate[44]. After decellularization, myogenic cells FACS isolated from new born wild type mice (CD45$^-$;CD31$^-$;Ter119$^-$; Sca1$^-$;α7-integrin$^+$) were seeded on the ECM substrate and tested for proliferation. ECM produced by Osr1$^{GCE/+}$ cells significantly enhanced myogenic cell proliferation in comparison to cultivation on ECM produced by Osr1$^{GCE/GCE}$ cells. (Fig. 9c). As a model for an isolated ECM defect we analysed the number of myogenic cells in *Col6a1* mutants, in which production of mature COL6 is disrupted[45]. *Col6a1* mutants showed a comparable decrease in myogenic cell number as *Osr1* mutants in E11.5 limbs (Fig. 9d). Together this suggests that Osr1$^+$ cells promote the expansion of the myogenic progenitor pool via the ECM, with COL6 as a major constituent and direct Osr1 target.

It was shown before that neonatal limb MCT fibroblasts supported myogenesis in transwell assays[12] indicating that MCT cells exert at least part of their function via the production of secreted signalling molecules. We discovered several secreted signalling factors downregulated in Osr1-deficient embryonic FAPs, amongst them Cxcl12 as direct Osr1 target. Cxcl12 and its receptor Cxcr4 are known to be involved in muscle cell proliferation in vitro[46, 47], and both Cxcl12 and Cxcr4 are involved in mouse limb myogenesis in vivo[48, 49]. We tested if Osr1$^+$ cells have the ability to enhance myogenesis via diffusible components using transwell assays. FACS-isolated E13.5 Osr1$^{GCE/+}$ or Osr1$^{GCE/GCE}$ cells were seeded into transwell inserts, myoblasts FACS isolated from new born wild type mice were seeded into cultivation wells. Osr1$^{GCE/+}$ cells significantly supported myogenesis as compared to cultivation with Osr1$^{GCE/GCE}$ cells or cultivation without any cells (Fig. 9e). To further elucidate to which extent the downregulation of the direct Osr1 target *Cxcl12* may be involved in the Osr1 KO phenotype we

analysed mouse mutants for the Cxcl12 receptor Cxcr4 at E14.5 for muscle patterning defects via whole limb immunolabelling. We observed defects that partly overlap with those of *Osr1* mutants, such as reduction on muscle size in the Biceps femoris muscles (Fig. 9f, g). For the Biceps femoris anterior head the phenotype of *Cxcr4* mutants was exacerbated compared to *Osr1* mutants (Fig. 9g). In C2C12 myogenic cells, the Cxcl12/Cxcr4 axis signals via the MAPK/ERK pathway[47]. The ERK pathway is also one major downstream component of integrin/focal adhesion signalling triggered by cell-ECM interaction in myogenesis[50]. ERK phosphorylation was significantly decreased in Lbx1$^+$ myogenic progenitors in *Osr1* mutants (Fig. 7h). A downstream target of ERK in this context is Cyclin D1[51, 52], which was less abundant in Lbx1$^+$ nuclei in *Osr1* mutants compared to wildtype (Fig. 7i). Altogether, we propose that Osr1$^+$ embryonal FAPs positively regulate myoblast proliferation and survival via a combination of ECM and paracrine signalling.

## Discussion

Correct number and distribution of myogenic progenitors emanating from the initial muscle anlagen is essential for embryonic muscle patterning, a process controlled by local MCT cells[8–10]. During this process the MCT cells are thought to create a promyogenic environment[13], such as a myogenic niche, that is so far not well characterized. We show here that Osr1$^+$ cells define a population of embryonic FAPs as a subpopulation of MCT. We demonstrate in vitro that Osr1$^+$ cells are capable of creating such a niche environment and that Osr1 is essential for their promyogenic capacity in vitro and in vivo.

Our data suggest that Osr1 specifies MCT cell fate and drives a transcriptional program of MCT-ECM genes in these cells. Thereby Osr1 directly controls the expression of *Col6a1* and *Col6a3*, encoding two essential subunits of COL6. The ECM produced by Osr1$^+$ cells provides an environment for myogenic cells that promotes their proliferation and survival. Based on the transcriptome data, we propose that Osr1 drives the transcription of ECM genes, directly or indirectly, that includes not only a subset of ECM components such as Collagens, but also components essential for collagenous matrix assembly as Lumican, Matrillin, Decorin and Fibromodulin[53]. Hence the combined defect seen in *Osr1* mutants should exacerbate effects seen in mouse models for individual ECM components such as *Col6a1* null mice. Indeed, the number of myogenic cells displays a more pronounced decrease in Osr1$^{GCE/GCE}$ than in *Col6a1*$^{-/-}$ embryos at E11.5. In addition, no obvious muscle patterning phenotype was observed in 3D reconstructions of *Col6a1*$^{-/-}$ limbs at E14.5 (Supplementary Fig. 11), indicating that myogenic cell numbers may be caught up later in *Col6a1*$^{-/-}$ embryos. This is not the case in *Osr1* mutants, where the number and proliferation rate of

---

**Fig. 9** *Osr1*-deficiency impairs the myogenic niche in developing limbs. **a** Decreased abundance of COL6 and FN (Fibronectin) in proximal (inserts 1 and 3) and dorsal (inserts 2 and 4) muscle primordia of *Osr1*-deficient E11.5 limbs. Boxed regions are shown in the numbered panels to the right and are magnifications of the area. **b** Reduced abundance of interstitial FN and disarrangement of COL14 in E13.5 Osr1$^{GCE/GCE}$ limb muscles as compared to wild-type littermate controls. **c** Extracellular matrix (ECM) produced by Osr1$^{GCE/+}$ cells, but not ECM produced by Osr1$^{GCE/GCE}$ cells supports myoblast proliferation (n = 3). **d** E11.5 *Col6a1*$^{-/-}$ embryos show a decrease in myogenic progenitor numbers (n = 3). **e** Transwell assay shows a beneficial effect of E13.5 Osr1$^{GCE/+}$ embryonic FAPs (eFAPs) but not Osr1$^{GCE/GCE}$ eFAPs on myogenesis. Transwell culture without eFAPs was used as control (n = 3). **f** *Cxcr4*$^{-/-}$ embryos show muscle defects overlapping with Osr1$^{GCE/GCE}$ embryos. **g** Sections of *Cxcr4*$^{+/+}$ and *Cxcr4*$^{-/-}$ embryos show almost complete reduction of the biceps femoris anterior head and hypoplasia of the biceps femoris accessory head. **h** Decreased phosphorylated ERK (pERK) and (**i**) CyclinD1 in myogenic Lbx1$^+$ cells in E11.5 *Osr1* mutants (n = 3). **j** Schematic model of Osr1$^+$ cell function. Osr1$^+$ cells (green) produce chemokines as Cxcl12 and a specific extracellular matrix (ECM) supporting myoblast proliferation and survival involving pERK signaling. Cxcl12 in addition likely provides local migratory cues ensuring correct spatial distribution of myoblasts as a prerequisite for muscle patterning. **k** Schematic model of limb muscle connective tissue compartmentalization. Specific transcription factors (Osr1 and Tcf4 are shown as examples) are expressed regionally in the limb bud with areas of overlap creating a regionalized code that likely determines the composition of the local niche. Error bars represent s.e.m. T-test: *=$p < 0.05$; **=$p < 0.01$; ***=$p < 0.001$. N-numbers indicate biological replicates, i.e. samples from different specimen in **d**, **h**, **i** or independent assays in **c**, **e**

myogenic cells are still decreased at E13.5 (Supplementary Fig. 8e, f) and patterning defects are already visible at this stage (Fig. 5d).

In addition to providing a beneficial ECM environment, we show that Osr1[+] embryonic FAPs express signalling molecules such as Cxcl12 and Bmp4 known to be involved in myogenesis. Both factors can promote myoblast proliferation[46, 47, 54, 55]. Cxcl12/Cxcr4 signalling was also involved in myoblast survival[48] thus defective signalling likely contributes to the reduction of the myogenic pool on top of the ECM defects described above. Intriguingly, both ECM / focal adhesion signalling as well as Cxcl12/Cxcr4 signalling can feed into the ERK pathway[47, 50], which we show is strongly impaired in limb myoblasts of Osr1[−/−] embryos. Moreover, Cxcl12 promotes myoblast migration in vitro and in vivo[48, 56] and deficiency of Cxcr4 impairs myoblast migration in vivo[48]. A significant early migration phenotype was however only observed in compound mutants for Cxcr4 and Gab1, an adaptor and signal mediator protein for c-met[48]. An overt muscle patterning phenotype was not demonstrated in Cxcr4 mutant mice, however they show a mild reduction of myogenic cell numbers in the limb with increased apoptosis and altered distribution of myoblasts[48]. We show here that Cxcr4 mutants show overlapping muscle patterning defects as Osr1 mutants. In case of the Biceps femoris anterior head Cxcr4 mutants show increased severity compared to Osr1 mutants, which could be expected from a full loss of a receptor as compared to a partial loss of its ligand. We propose that Cxcl12 produced by Osr1[+] cells may provide local migration cues for myogenic cells to reach their correct target areas in the phase of muscle patterning. A failure of this local mechanism may explain the absence of myogenic cells in areas normally populated by myoblasts (Fig. 5b). We therefore conclude that embryonic Osr1[+] FAPs are a vital part of a developmental myogenic niche consisting of a muscle-specific ECM and a favourable signalling environment. Both components are linked, since composition of the ECM strongly influences signalling by secreted molecules[57] and both mechanisms can feed into overlapping intracellular signalling pathways that together maintain the myogenic cell pool and ensure its correct spatial distribution via directed local migration (Fig. 9j).

Several transcription factors that are expressed in limb connective tissue were shown to influence muscle patterning, namely Tcf4, Tbx4, Tbx5, Hoxa11 and Hoxd11[12, 58, 59], although their modes of action are mostly unclear. None of these genes were obviously deregulated in Osr1-deficient cells or in Osr1-deficient limbs (Supplementary Fig. 9d, f) suggesting that Osr1 acts in a different genetic pathway or downstream of these transcription factors.

To date, Tcf4 is the most commonly accepted marker for MCT fibroblasts. We show that during embryonic myogenesis, Tcf4 exhibits a highly regionalized expression, which is mutually exclusive with Osr1 in many places and overlapping in others (Fig. 2). This implies that neither Tcf4 nor Osr1 are universal markers for limb MCT. However, the level of coexpression of Osr1 and Tcf4 in MCT cells is increased during development, and Tcf4[+] cells can arise from Osr1[+] progenitors. This is in accordance with the uniform expression of Tcf4 in MCT of all limb muscles in neonatal mice as reported by Mathew et al.[12]. This suggests that embryonic MCT has a regionalized character where subpopulations are identifiable by the expression of specific transcription factors. During development each positional value in the limb may be characterized by a combination of instructive transcription factors. This concept suggests a toolbox of transcription factors as easily adaptable determinants of the specific local myogenic niche character, which might control the shaping of individual muscles (Fig. 9k). After the initial patterning phase this regional MCT subdivision seems to dissolve since at birth most MCT cells appear to have assumed Tcf4 expression[12], while Osr1 expression is fading.

We show here that the embryonic FAPs are lineage progenitors of adult muscle interstitial FAPs. Osr1[+] cells uniquely contribute to FAPs and not to other mononuclear cell populations identifiable by the FACS protocol we used[20]. The overall contribution of developmental Osr1[+] cells to adult limb FAPs detected by our protocol was moderate. This is however not surprising given that for cell isolation all limb muscles were used, while Osr1 is only expressed in a subset of muscles during development. Consequently, when counting the contribution of Osr1[+] progenitors to FAPs on sections, a considerably higher contribution rate was found in individual muscles correlating with developmental Osr1 expression. We chose a protocol of timely limited administration of moderate TAM doses over either a time-extended administration of high TAM doses or constitutive Cre strategies, since our strategy provides high specificity, however at the expense of efficacy. Together, our data show that adult FAPs derive in part from Osr1[+] progenitors, but this is unlikely the only developmental source. Of note we report here quantitative data showing that a majority of adult FAPs express Tcf4 (Fig. 4d) in agreement with observations made by Murphy et al.[14] suggesting that Tcf4[+] muscle connective tissue fibroblasts and FAPs are in fact largely, however not completely overlapping populations in the adult muscle interstitium, as proposed previously[14, 19]. During muscle regeneration, FAPs are thought to be vitally involved in creating a specific, yet mostly undefined, niche for satellite cells[14, 17, 19, 20, 60]. Future studies will determine whether adult FAPs employ similar mechanisms for regenerative myogenesis as their developmental predecessors.

## Methods

**Animals**. Mice were maintained in an enclosed, pathogen-free facility, and experiments were performed in accordance with European Union regulations and under permission from the Landesamt für Gesundheit und Soziales (LaGeSo) Berlin, Germany (Permission numbers ZH120, G0346/13, G0240/11). Mouse lines used: Pax3[GFP61], Osr1[GCE25], R26R[mTmG62], R26R[LacZ63], Col6a1[+/−45] and Cxcr4[+/−64].

**Cell proliferation analysis**. Bromodeoxyuridine (BrdU, Roche) was administered intraperitoneally to pregnant females (50 mg kg[−1]) at the desired stage. Embryos were collected 1 h after BrdU administration for assessing proliferation rates. After immunolabelling (see below), Lbx1[+], Pax7[+] or Myod1[+] nuclei positive or negative for BrdU, respectively, were counted in a defined area. To establish the proliferation rate double positive cells were set in relation to the total cell number positive for the respective marker.

**Tamoxifen administration and Osr1 cell lineage tracing**. Tamoxifen (Sigma Aldrich) was dissolved in a 1:10 ethanol/sunflower oil mixture. For embryonic lineage tracing, pregnant R26R[mTmG/mTmG] females that had been bred to Osr1[GCE/+] males were injected with 150 µl of a 20 mg mL[−1] Tamoxifen stock. Tissues were collected at E18.5. Upon CreERt2 activation, the mTmG allele is converted as to switch cells from membrane tomato (mT) expression to membrane eGFP (mG) expression. Importantly, mG expression driven by the CMV enhancer/beta-actin core promoter cassette can be visualized by direct microscopy without antibody labelling. This, and the membrane localization, differentiates mG expressed from the R26R[mTmG] allele from the cytoplasmic eGFP expressed from the Osr1[GCE] allele, which was not detectable without prior immunolabelling for GFP under the experimental conditions used here.

For postnatal lineage tracing, new born Osr1[GCE/+];R26R[LacZ/+] or Osr1[GCE/+]; R26R[mTmG/+] pups were injected subcutaneously into the neck fold with 25 µl of a 3 mg ml[−1] Tamoxifen stock. Adult tissue was collected at 12 weeks of age. Expression of beta-Galactosidase (bGal) from the R26R[LacZ] allele was detected by immunolabelling (see below).

**Tissue preparation**. For immunolabelling, embryonic tissues were fixed in 4% PFA for 2 h at RT and treated with successive 15 and 30% (w/v) sucrose (Roth) solutions before O.C.T. (Sakura) cryo-embedding in a chilled ethanol bath. Embryonic tissue was sectioned at 12 µm thickness and stored at −80 °C until use. Adult muscle tissues were directly embedded in gum Tragacanth 6% (w/v) (Sigma-Aldrich) and flash frozen using a chilled (−160 °C) isopentane (Carl Roth) bath. Adult tissues were cut at 7 µm thickness and fixed in 4% PFA for 5 min at room temperature. For section in situ hybridization tissue was fixed in 4%

paraformaldehyde dissolved in diethylpyrocarbonate (DEPC)-treated PBS for 2 h at room temperature. Tissue was sectioned at 10 μm thickness and stored at −80 °C until use. For whole-mount in situ hybridization embryos were fixed in 4% PFA/DEPC-PBS at 4 °C overnight. After fixation embryos were washed with DEPC-PBS and 0,1% Tween-20 (PBST) for 30 min on ice. Embryos were dehydrated in Methanol and stored at −20 °C until use.

**Immunolabelling.** Permeabilization of sections was performed in 0.3% (v/v) Triton X-100 (Sigma Aldrich) in phosphate buffer (PBS) for 10 min. Sections from adult tissues were blocked with 5% (w/v) bovine serum albumin (Sigma Aldrich) in PBS. Antigen retrieval was performed where necessary: for Pax7 immunostaining, sections were treated for 6 min in −20 °C chilled methanol and subsequently washed in PBS. Further epitope retrieval was performed by a treatment with 1 mM Ethylendiaminetetraacetic acid (Roth) at 95 °C for 10 min. For BrdU detection sections were treated using 7.5% 1 N HCL at 37 °C during 30 min. Cleaved Caspase 3 antibody staining was preceded by an antigen retrieval consisting of a treatment with citrate buffer, 10 mM sodium citrate (Roth) pH 6, at 95 °C for 10 min. Sections were blocked with 5% horse serum, 5 mg mL$^{-1}$ blocking reagent (Perkin Elmer) and 0,1% Triton X-100 in PBS for 1 h at RT. Primary antibodies were incubated at 4 °C overnight, followed by secondary antibody staining of 1 h at room temperature. Antibodies used for these experiments are listed in Supplementary Tables 3 and 4. Specimens were counterstained with 5 μg μL$^{-1}$ 4′,6-diamidino-2-phenylindole (DAPI; Invitrogen) and mounted with FluoromountG (SouthernBiotech).

For immunolabelling of cells samples were fixed using 4% PFA for 2 h at room temperature. Samples were blocked and permeabilized with 5% horse serum, 5 mg mL$^{-1}$ blocking reagent (Perkin Elmer) and 0.1% Triton X-100 in PBS for 1 h at RT followed by primary and secondary antibody incubation as above. Cytospin-collected cells were fixed with 4% PFA for 10 min at 4 °C and subjected to immunolabelling as above.

**In situ hybridization.** Digoxygenin-labelled Riboprobes for in-situ hybridization were made with the DIG RNA labelling kit (Roche: 11175033910) according to the manufacturer's instructions. Probes used in this study are listed in Supplementary Table 5.

For section in-situ hybridization cryosections were washed three times with DEPC-PBS and treated with acetylation buffer (0,75% triethanolamine, 0,15% hydrogen chloride and 0,2% acetic anhydride in DEPC-H$_2$O) for 10 min under agitation. Acetylation buffer was washed out with DEPC-PBS. Sections were incubated for at least 2 h at room temperature in pre-hybridization buffer (50% Formamide, 5x concentrate sodium saline citrate buffer (SSC) pH 4.5, 1% SDS, 0,1% Tween-20). Riboprobes were diluted in hybridization buffer (10 mM Tris pH 7.5, 600 mM NaCl, 1 mM EDTA, 0,25% SDS, 10% Dextran sulfate, 1x Denhardt´s, 200 μg/ml yeast tRNA and 50% Formamide) and added to the slides. Slides were covered with a glass coverslip followed by hybridization at 65 °C overnight in a humidified chamber. Coverslips were removed by washing in 2x SSC for 5 min at room temperature. Unhybridized RNA probe was removed by washing the slides twice in 2x SSC at 65 °C for 30 min followed by two washing steps in MABT-buffer (100 mM Maleic acid pH 6, 0.1% Tween) for 30 min at room temperature. Slides were blocked in 500 μl blocking solution (MABT-buffer + 5% blocking reagent Roche 1109617001) for 1 h at room temperature in a PBS humidified chamber. Anti-DIG antibody (Roche: 11093274910) was dissolved 1:2000 in blocking solution, added to the sections and incubated overnight at room temperature. Antibody solution was washed off with MABT five times for 30 min at room temperature. Slides were treated twice with NTMT-buffer (100 mM NaCl, 100 mM Tris pH 9.5, 50 mM MgCl$_2$ and 0,1% Tween-20) for 10 min at room temperature followed by signal detection with detection solution (5 μg 4-Nitro blue tetrazolium chloride (NBT) and 2.6 μg 5−bromo-4-chloro-3-indolyl-phosphate (BCIP) dissolved in NTMT). Signal detection was performed at 37 °C in a light protected chamber. Staining reaction was stopped by washing the slides in PBS for 5 min. If further antibody staining was desired, sections were directly processed for immunolabelling (see above).

For whole-mount in-situ hybridization embryos were rehydrated in DEPC-PBS and bleached in 6% hydrogen peroxide in PBST for 1 h on ice. For permeabilization embryos were treated with 20 mg/ml proteinase K at room temperature for 8 min. Embryos were washed twice in DEPC-PBST and post-fixed for 20 min in 4% PFA in DEPC-PBS at room temperature followed by brief washing in DEPC-PBST. For pre-hybridization embryos were incubated in L1−Buffer (50% Formamide, 5x SSC pH 4.5, 1% SDS and 0,1% Tween-20) for 10 min at 68 °C. Then embryos were incubated in H1-Buffer (L1-Buffer supplemented with 0,1 mg/ml tRNA and 0,05 mg/ml Heparin) for 2 h at 68 °C. Riboprobes were diluted in H1-Buffer and added to embryos in 2 ml cryovials followed by hybridisation at 68 °C overnight. Unbound probe was removed by washing the embryos 3 times for 30 min in 68 °C in pre-warmed L1-Buffer followed by 3 times washing in L2-Buffer (50% Formamid, 2x SSC pH 4.5, 0,1% Tween-20 in DEPC-water) for 30 min at 68 °C. Embryos were then washed for 15 min at 68 °C with L3-Buffer (L3-Buffer: 2x SSC pH 4.5, 0,1% Tween-20 in DEPC water) followed by washing in a 1:1 solution of L3-Buffer and RNase buffer (500 mM NaCl, 10 mM Tris pH 7.5, 0,1% Tween-20) for 5 min at room temperature. Embryos were treated with a solution of 100 μg/ml RNase A in RNase buffer for 1 h at 37 °C. RNase solution was washed out 3 times for 5 min each at room temperature with Tris saline buffer (8 g/L NaCl, 0,2 g/L

KCl, 3 g/L Tris pH 7.4) supplemented with 1% Tween-20. Embryos were blocked in blocking solution (2% fetal calf serum, 2 mg/ml bovine serum albumin dissolved in TBST) at room temperature for 2 h. Anti-DIG antibody (Roche: 11093274910) was dissolved 1:5000 in blocking solution and incubated overnight under rotation at 4 °C. Embryos were washed in TBST supplemented with alkaline phosphatase inhibitors 0,05% Levamisole/Tetramisole (Sigma Aldrich L9756) 5 times for 5 min each, and again 8 times for 30 min each. Embryos were then incubated in alkaline phosphatase buffer (100 μM NaCl, 50 mM MgCl$_2$, 0,1% Tween-20, 100 mM Tris pH 9.5 and 0,5 mg/ml Levamisole/Tetramisole) for 20 min. Signal detection was performed with 225 μg/ml NBT and 87,5 μg/ml BCIP dissolved in alkaline phosphate buffer at room temperature under light protection. Signal development was stopped by washing for 3 times 10 min each in alkaline phosphate buffer followed by fixation in 4% PFA, 0,2% glutaraldehyde and 5 mM EDTA for long term storage.

**Cell isolation and flow cytometry.** Limbs of E13.5 $Osr1^{GCE/+}$ or $Osr1^{GCE/GCE}$ embryos were minced in 500 μL high-glucose Dulbecco's modified eagle medium (DMEM, Invitrogen) using a small scissor. Further enzymatic digestion of the tissue was performed using 10 mg ml$^{-1}$ of Collagenase (Collagenase NB 4 G proved grade, SERVA Electrophoresis$^{TM}$) in DMEM medium supplemented with 2 mM CaCl$_2$ and MgCl$_2$ with vigorous shaking at 37 °C for 45 min. After digestion, tissue extracts were washed and resuspended in high-glucose DMEM medium. Cell suspensions were passed through a 100-μm cell strainer (Miltenyl Biotec) and collected by centrifugation at 300 g for 5 min. For flow cytometry analysis or cell sorting, cell suspensions were blocked for 5 min. in 5% mouse serum (Sigma Aldrich). Antibody labelling (antibodies see Supplementary Table 3) was performed for 60 min at 4 °C. Before flow cytometry cell suspensions were washed, collected by centrifugation at 300 g for 5 min and passed through a 35-μm cell strainer filter (BD Biosciences). To assess viability, cells were stained with propidium iodide (1 μg ml$^{-1}$, eBioscience) immediately before sorting or analysis.

For cell isolation from adult muscles whole hind limb muscles were carefully isolated, roughly minced and digested in high-glucose DMEM medium containing 2,5 mg/ml Collagenase A (Roche) for 75 min at 37 °C with vigorous shaking. 2 IU/ml of Dispase II (Sigma Aldrich) were added to the digestion solution and muscle lysates were digested for further 30 min. Muscle slurries were passed 10 times through a 20 G syringe (BD Bioscience) and a 70-μm cell strainer. Cells were collected by centrifugation at 300 g for 5 min and resuspended in staining buffer consisting of 500 μl Hank´s balanced salt solution (HBSS, Thermo Fisher scientific), 0.4% bovine serum albumin (Sigma Aldrich) and 20 μg/ml Gentamycin (Serva Electrophoresis). Cells were labelled with antibodies (Supplementary Table 6) on ice for 30 min and washed twice with staining buffer previous to FACS sorting. Propidium iodide was used as a viability dye.

Sorts and analyses were performed on a FACS Aria II (BD Biosciences). Data were collected using FACSDIVA software. Biexponential analyses were performed using FlowJo 10 (FlowJo LLC) software. Analysis was performed on three independent biological replicates (i.e. cells FACS isolated from three different embryos). Sorting gates were defined based on unstained controls. Cells were sorted either into 1 ml high-glucose DMEM warmed to 37 °C for further culturing or directly into buffer RLT (Qiagen RNeasy kit) for RNA extraction.

**Cell culture.** FACS isolated cells were collected by centrifugation at 300 g for 5 min and seeded on 2% Matrigel$^{TM}$ (BD Biosciences) coated glass coverslips (Thermo scientific) previously exposed to 30 mins of UV light. Before differentiation induction cells were then expanded for 3 days in growth medium (high-glucose DMEM medium with 10% fetal bovine serum (FBS, Biochrom)). FACS isolated cells designated for adipogenic differentiation were first expanded for 3 days in medium containing 60% low-glucose DMEM (Invitrogen), 40% MCDB201 Media (Sigma Aldrich), 10% FBS, 0.4 ng mL$^{-1}$ dexamethasone, 29 ng mL$^{-1}$ L-ascorbic acid 2-phosphate, 1x ITS liquid media supplemented mix, 1x Linoleic acid-albumin (Sigma Aldrich) and freshly added growth factors: 10 ng mL$^{-1}$ EGF, 10 ng mL$^{-1}$ PDGF-BB (PeproTech), 10 ng mL$^{-1}$ LIF (Millipore) and 5 ng mL$^{-1}$ bFGF (Sigma Aldrich).

Myogenic differentiation was induced by culturing cells for 5–7 days in high-glucose DMEM medium with 2% horse serum (Vector Labs). For fibrogenic differentiation cells were cultured in low-glucose DMEM medium with 2% FBS supplemented with 1 ng mL$^{-1}$ TGFβ1 (PeproTech) for 3 days followed by 3 days in differentiation medium without TGFβ1. For osteogenic differentiation, we used low-glucose DMEM medium supplemented with 100 nM dexamethasone, 10 mM β-glycerophosphate, 0.2 mM ascorbic acid and 50 ng ml$^{-1}$ L-thyroxine (Sigma Aldrich) and differentiated the cells for 40 days with medium change every two days. For chondrogenic differentiation we seeded 20 μl droplets of $10^5$ cell suspensions on uncoated coverslips. Cultures were grown in high-glucose DMEM medium with 10% FBS and supplemented with 10 ng mL$^{-1}$ TGFβ1, 1 μM ascorbic acid, 100 nM dexamethasone and 10x ITS liquid media supplemented mix (Sigma Aldrich) for 14 days. Medium was changed every 2–4 days. For adipogenic differentiation cells were cultured for 48 h in 60% low-glucose DMEM (Invitrogen), 40% MCDB201 Media (Sigma Aldrich), 10% FBS supplemented with 1 μM dexamethasone, 0.5 μM isobutylmethylxanthine, 5 μg mL$^{-1}$, 50 μM indomethacin, 1 nM triiodo-L-thyronine sodium salt (Sigma Aldrich) and insulin (Roche). Thereafter cells were cultured for 7 days in 60% low-glucose DMEM (Invitrogen),

40% MCDB201 (Sigma Aldrich), 10% FBS. Differentiation assays were performed on three independent biological replicates (i.e. cells FACS isolated from three different embryos).

ECM deposition assay was conducted as follows: Osr$^+$ cells were isolated via FACS sorting from E13.5 Osr1$^{GCE/+}$ or Osr1$^{GCE/GCE}$ embryos and plated on 24 well-plates (52.500 cells/well). Cells were kept in growth medium (10% FCS, DMEM 4,5 g Glucose, 1% Penicillin/Streptomycin, 1% L-Glutamine) until they reached 80% confluence. Cells were further cultured in low serum medium (2% FCS, DMEM 4,5 g Glucose, 1% Penicillin/Streptomycin, 1% L-Glutamine) for 14 days. Decellularization was conducted via a snap freeze/thaw procedure. Cells were washed once with H$_2$O, then wells were filled with 200 µl of H$_2$O and plates were stored for 40 min at −80 °C. Plates were thawed in a 37 °C water bath for 30 s. Then the wells were filled with 800 µl growth medium (room temperature) and transferred to 37 °C. Myoblasts were collected from 3–4 days old neonate mice via FACS (CD45$^-$;CD31$^-$;Ter119$^-$;Sca1$^-$;α7-integrin$^+$). Myoblasts were seeded on the ECM (2,5 × 10$^5$ cells/well). As control Myoblasts were seeded in wells coated with 10% Matrigel (Corning). Myoblasts were then cultured in proliferation medium (20% FCS, 10% horse serum, DMEM 4,5 g Glucose, 1% Penicillin/Streptomycin, 1% L-Glutamine) for 16 h. Then cells were fixed with 4% PFA and immunolabeled for Pax7 and Ki67.

Transwell assay was performed as follows: 2,5 × 10$^5$ FACS-isolated p3–4 myoblasts (see above) were seeded on glass coverslips coated with Matrigel$^{TM}$ and allowed to settle for 24 h in myogenic proliferation medium (DMEM 4,5 g Glucose, 1% Penicillin/Streptomycin, 1% L-Glutamine, 10% horse serum, 20% FBS) in 24 well plates. The day before Osr1$^+$ cells were isolated via FACS from E13.5 Osr1$^{GCE/+}$ or Osr1$^{GCE/GCE}$ embryos. 8 × 10$^4$ GFP$^+$ cells were seeded into transwell inserts and cultured for 24 h in growth medium. On the next day, growth medium was replaced by myogenic differentiation medium (DMEM 4,5 g Glucose, 1% Penicillin/Streptomycin, 1% L-Glutamine, 2% horse serum) for preconditioning. After 24 h the transwell inserts containing the GFP$^+$ cells together with the preconditioned myogenic differentiation medium were transferred to 24 well plates containing the myoblasts. As control, myoblasts were cultured with empty transwell inserts. Transwell coculture was performed for 14 days with medium change every second day. Myogenic differentiation was analyzed by immunolabeling for Myosin heavy chain (MyHC). For quantification an area of 1,75 mm$^2$ were analyzed and only MyHC$^+$DAPI$^+$ cells were counted.

**Staining procedures**. To stain lipids, cell cultures were stained with Oil Red O (Sigma Aldrich) 0.5% (w/v) 1 h at RT. Lipids on sections were stained 15 min in Oil Red O staining solution at RT. Calcium accumulation after osteogenic differentiation was assessed by Alizarin Red (Sigma Aldrich) staining using a 1% (v/v) staining solution (from a stock solution of 13.3 mg mL$^{-1}$ at pH 6.5) applied during 30 min. For assessing chondrogenic differentiation cultures were stained using Alcian Blue (Sigma Aldrich) staining solution 10 mg mL$^{-1}$ dissolved in 0.1 N hydrochloric acid (Carl Roth) applied during 30 min at RT.

**Imaging**. Cell cultures were documented with a Nikon Eclipse TS 100 microscope, a Leica DMR microscope or a Zeiss SteREO Discovery V12 stereomicroscope. Confocal images of immunolabelled sections or cells were taken using the confocal laser scanning microscope system LSM710 (Zeiss). Images were captured using Axio Vision Rel. 4.8 and Zen 2010 (Zeiss).

**Cell quantification**. Quantification of FACS-isolated Osr1$^+$ cells after immunolabelling (cytospin or growth medium culture) was performed on three independent biological replicates (i.e. cells FACS isolated from three different embryos). Quantification of specific cell markers on sections after immunolabelling was assessed on at least 4 serial sections in 30 µm intervals for each biological replicate with a minimum of 3 replicates (exact values given in figures, in each case n stands for one biological replicate). In Fig. 3g the interval was reduced to 10 µm. For each experiment serial sections of the whole limb were immunolabelled, and only sections containing comparable regions were used for quantification. At least five sections for each biological replicate were counted. Student's t-test was performed using Prism 5 (GraphPad) software. Error bars in all figures, including supplementary information, represent the mean ± standard error of the mean (s.e.m).

**3D reconstruction of embryonic limb muscles**. Whole mount immunofluorescence labelling on E14.5 embryonic limbs was performed as previously described[65]. For 3D reconstruction, images captured on a LSM710 confocal microscope (Zeiss) were analysed using the programs Amira® (FEI visualization Sciences Group) and Fluorender® (University of Utah).

**Quantitative real-time PCR**. RNA isolation was performed using RNeasy mini kits (Qiagen), and RNA quantification was assessed using a ND2000 spectrophotometer (Nanodrop). Reverse transcription was performed using the High Capacity cDNA Reverse Transcription Kit (Applied Biosystems). Gene expression analysis was conducted using Taqman Gene Expression Assays (Applied

Biosystems), on a 7900HT Real Time PCR system (Applied Biosystems). Primer sequence information is provided in Supplementary Table 7. Data were acquired and analysed using SDS 2.0 software (Applied Biosystems). Quantitative RT-PCR analyses were performed on at least four independent biological samples (embryos) different from those used for RNA-Sequencing.

**RNA-sequencing**. Total RNAs were extracted from Osr1$^+$ cells isolated by FACS-sorting from E13.5 limbs of two Osr1$^{GCE/+}$ and two Osr1$^{GCE/GCE}$ embryos by using the RNeasy mini kit (Qiagen). Libraries were prepared by using the TruSeq RNA Library Preparation Kit v2 according to Illumina's intructions and subjected to high-throughput sequencing by using the Illumina HiSeq 2500 technology. 65–75 million of 50-bp paired-end reads were generated with a mean insert size of 150 bp and mapped against the genome of *Mus musculus* version mm9 by using TopHat2[66] and the UCSC gene model annotation from iGenome as guide (default parameters, except for the mean inner distance set at 150 and the minimum intron length set at 50). Aligned fragments per gene were counted by using the intersection strict mode from HTSeq[67]. Raw fragment counts showed a very high Pearson correlation (>0.99) between both biological replicates for each condition. Differential expression analysis was performed by using DESeq2[68] and a false discovery rate (FDR; alpha) of 0.05. Genes were considered as being differentially expressed between the Osr1$^{GCE/+}$ and Osr1$^{GCE/GCE}$ conditions if the Benjamini-Hochberg adjusted p-value (padj) was below 0.1. Transcripts per million (TPM) abundances were calculated from the mean normalized fragment counts given by DESeq2 for both Osr1$^{GCE/GCE}$ samples. Gene ontology (GO) analyses were carried out by using the DAVID functional annotation tools[69]. Raw fastq and count data were uploaded on the Gene Expression Omnibus (GEO) database under the accession number GSE78056.

**Chromatin immunoprecipitation**. Genomic regions up to 20 kb upstream of putative direct Osr1 targets were screened for consensus Osr1 binding sites in the UCSC browser (GRCm38/mm10). For Cxcl12, in addition the gene desert 5′ of the first exon was screened for Osr1 binding sites specifically within DNAseI hypersensitive regions (putative enhancers). Distant binding sites were confirmed to be included into the same topology-associated domain as Cxcl12 (http://promoter.bx. psu.edu/hi-c/view.php)[70] PCR primer were designed overlapping or closely adjacent to binding sites. Regions without consensus binding sites located in the vicinity were chosen as controls.

Lentiviral particles carrying an Osr1-tripleFLAG construct under the ubiquitin c (UbC) promoter or carrying a GFP expression construct as control were purchased from AMSBio (Abingdon, UK). For ChIP assays, MEFs were isolated from E13.5 C57Bl6 limbs as described above (Cell isolation). For infection 3 × 10$^5$–4 × 10$^5$ cells were resuspended in 5 ml 37 °C pre-warmed growth medium (high glucose DMEM medium supplemented with 10% FBS). Infection was achieved using spinoculation of suspension cells. Polybrene and lentiviral particles were added to the medium. Cells and virus particles were centrifuged for 60 min at 300 g before seeding. Medium was changed at the next day. Chromatin immunoprecipitation (ChIP) was performed as follows: cells were fixed on a confluent 10 cm dish for 10 min with 10 ml of 1% Formaldehyde (Sigma Aldrich) in growth medium. Chromatin cross-linking was quenched adding 550 µl of 2.5 M Glycine. Medium was removed and cells were washed twice with DPBS. Cells were lysed and removed with a scratcher. Nuclei were collected in 900 µl sonication buffer and split into 3 × 1.5 ml TPX tubes (Biogenade). Cells were sonicated for 20 cycles in a Bioruptor (Biogenade). 30 µg of sonicated chromatin was used per ChIP. 8 µg of anti-FLAG antibody was added to the 30 µg of sonicated chromatin and incubated under rotation overnight at 4 °C. 30 µl of protein G beads (Invitrogen) were added to the Chromatin-Antibody mixture and incubated under rotation at 4 °C overnight. Chromatin bound to the Dynabeads-antibody complex was isolated using vigorously shaking conditions at 65 °C for 30 min and further de-crosslinked overnight at 65 °C. Proteins bound to the chromatin were removed with a 60 min treatment at 55 °C of Proteinase K (500 mg/ml) and RNA with a 30 min treatment at 37 °C with RNase A (Sigma Aldrich). Genomic DNA from sonicated chromatin was used as an input DNA and DNA isolated after ChIP as pulldown DNA. Pulldown was evaluated by Gel electrophoresis and quantitative real-time PCR, with 3 independent pulldown experiments. Primer sequences are provided in Supplementary Table 8. Uncropped lanes from representative gel electrophoresis images can be found in Supplementary Fig. 12.

**Data availability**. All data supporting the findings of this study are available within the article and its supplementary information files or from the corresponding author upon reasonable request. Raw fastq and count data from RNA Sequencing experiments have been deposited in the Gene Expression Omnibus (GEO) database under accession code GSE78056.

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

## Acknowledgements

This work was funded by the German Research Foundation (DFG; grant GK1631), French-German University (UFA-DFH; grant CDFA-06-11), the Association Française contre les Myopathies (AFM 16826), and the Fondation pour la Recherche Médicale (FRM DEQ20140329500) as part of the MyoGrad International Research Training Group for Myology. This work was funded by a grant from the AFM to D.D. and S.S. (Grant No. 18626), a grant from the FRM (grant No. FDT20150532272) to M.O., and by the Focus Area DynAge of the Freie Universität Berlin. We gratefully acknowledge Andrew P. McMahon (Harvard Medical School, USA) and Andreas Kispert (Hannover Medical School, Germany) for providing Osr1$^{GCE}$ and R26$^{mTmG}$ mouse lines, and Carmen Birchmeier (Max Delbrueck Centre, Berlin, Germany) for Cxcr4$^{+/-}$ mice, antibodies and technical advice. We thank Tim J. Schultz (German Institute of Human Nutrition, Potsdam) for advice on cell differentiation assays. We thank Petra Knaus and Christian Hiepen (Institute for Chemistry and Biochemistry, Freie Universität Berlin) for advice on the ECM assay.

## Author contributions

Experiments were conceived and designed by S.S., D.D. and P.V.G. P.V.G. performed the majority of experiments and data collection. S.vH.S., J.S., V.K. and D.M.I. performed experiments and analysed data. S.T.B. and B.T. performed and supervised the RNA-Sequencing procedure and M.O. performed RNA Sequencing data analysis. S.H., F.R., K.H., G.S. and M.K. provided reagents and mouse mutants. Data analysis and interpretation was performed by P.V.G. and S.S. with involvement of G.M., D.S. and D.D. S.S., D.D. and D.S. wrote the manuscript.

## Additional information

**Competing interests:** The authors declare no competing financial interests.

