## [Peer Review File · Nature Communications]

Reviewers' Comments:

Reviewer #1 (Remarks to the Author)

Garcia et al provide an interesting but mechanistically not very deep description of the role of the zinc-finger transcription factor Osr1 for formation of skeletal muscles during limb development of mice. The authors found that Osr1 is expressed in a subset of muscle connective tissue cells (MCT) during development sharing characteristics of so-called fibroadipogenic progenitor cells (FAPs), which is another word for mesenchymal stem cells in skeletal muscle. Inactivation of Osr1 resulted in malformation of muscles in the limbs but did not affect distribution of Osr1+ cells. RNA-seq analysis revealed a strong up-regulation of tendon and cartilage genes in Osr1-cells, which corresponds to earlier findings in neural crest-derived mesenchyme of tongue muscles (Liu et al, PNAS, 2013). In parallel, the authors detected reduced concentration of mRNAs for collagen6 (at E11.5) and collagen 3 and collagen5 (at E13.5) as well as a decrease of transcripts of several chemokines in Osr1-deficient cells. The authors conclude that the lack of Osr1 impairs muscle progenitor cell proliferation and positioning without dramatic effects on limb muscle progenitor cell migration.

The principal observation that lack of Osr1 disturbs formation of individual skeletal muscles during mouse limb development is interesting. It also seems likely that the dysregulation of ECM genes and chemokines in Osr1-mutant mesenchymal cells in the limb contribute to skeletal muscle limb formation. However, the whole analysis remains rather superficial. It does not become clear which effects in the Osr1-mutant mesenchymal cells are primary or secondary and what the specific contribution of changes in Osr1-mutant mesenchymal cells to skeletal muscle malformation is. The authors propose that Osr1 identifies a subpopulation embryonic FAPs, which is certainly true, but a more comprehensive quantitative analysis that would define the relative contribution of Osr1+ cells to FAPs and the relation of different subpopulation of FAPs to each other is missing.

Specific comments

The authors detected several genes that were either up- (261 genes) or downregulated (250 genes) in Osr1 mutant cells. Some of the dysregulated genes will most likely contribute to the phenotype of Osr1 mutants. However, the authors did not make any efforts to evaluate the individual contribution of any of the dysregulated genes. Is the downregulation of collagen 6 the major culprit? Is there a role for any of the other downregulated collagens? What is the impact of the downregulation of chemokines? I do not ask for a comprehensive assessment of all changes but a more in-depth functional analysis is absolutely mandatory.

Moreover, no attempts were made to understand the role of Osr1 in the pro-myogenic transcriptional program that enables FAPs to control skeletal formation in the lab. Some genes were downregulated were other were upregulated. Therefore, it seems very likely that several transcriptional changes in Osr1 mutant cells are secondary events (e.g. via Sox9 and others). A ChiP-Seq experiment might help to identify some of the primary targets of Osr1.

The authors claim that the function of muscle connective tissue cells during embryonic development and of FAPs during adult muscle regeneration might be similar without providing a clear proof for this hypothesis. Since Osr1 is widely expressed throughout the intermediate mesoderm and activated in many other tissues, including the branchial arches and limb buds at later embryonic stages, it does not seem to specifically define the embryonic origin of FAPs but rather identifies a specific subset of muscle connective tissue cells required for correct skeletal muscle patterning of the limb. The proposed parallel to FAPs in adult mice is superficial and poorly substantiated, in particular since the authors did not even take a superficial glimpse on adult mice. Formally, it is possible that Osr1+ cells only generate a transient subpopulation of FAPs during

limb muscle formation.

It is not clear to what degree *Osr1*⁺ cells contribute to the FAP/MSC population in different adult muscles. Do *Osr1*⁺ cells contribute to FAPs in other adult muscles? (If they really contribute to a major degree to adult FAPs). Is only the patterning of limb muscles affected in *Osr1* mutants?

The authors convincingly identify *Osr1* as a marker of a subpopulation of MCT cells in the embryonic limb and found a partial overlap with *Tcf4* expression, a popular marker of MCT cells in the developing limb. They also analyzed the expression of several typical markers of mesenchymal stem cells such *PDGFRalpha*. However, the relative input of *Osr1*⁺ cells to MCT in the limb remains enigmatic. It should be relatively easy to determine the overall contribution of *Osr1*⁺ cells to *PDGFRalpha* cells (FACS, co-staining on sections) in comparison to *Tcf4* and to other markers. The co-staining of *Osr1* with *Tcf4* needs to be done in both directions to see whether all *Tcf4* cells express *Osr1* or not. The staining for *Tcf4* on FACS-sorted *Osr1* cells does not provide this information. Are *Osr1*⁺/*Tcf4*⁺, *Osr1*⁺/*Tcf4*⁻ and *Osr1*⁻/*Tcf4*⁺ cells homogeneously distributed in the muscle anlagen of the developing limb? Such approaches would help to substantiate the hypothesis that a combination of instructive transcription factors in the MCT determines the local character of individual muscle anlagen.

The authors noted reduced presence and reduced proliferation of muscle progenitor cells in *Osr1* mutant, which lead to the hypothesis that *OSR1*⁺ FAPs directly stimulate proliferation of muscle progenitor cells. The authors hypothesize that the reduced expression of chemokines by *Osr1* mutant FAPs might be responsible for this phenomenon. However, chemokines have been mostly associated with migration of muscle progenitor cells. The effects on proliferation are usually rather weak. It also seems possible that the aberrant distribution of muscle progenitor cells in *Osr1* mutant limbs or a disturbed ECM composition contributes to reduced proliferation.

Due to the embryonic lethality of germ line *Osr1* mutants it is difficult to explore long-term effects and to verify the loss of individual muscles in newborn mice. Obviously, it would be much better to inactivate the *Osr1* gene with *PDGFRalpha*-Cre mice, although such an experiment requires substantial efforts and takes a long time. Nevertheless, such an approach would also validate a cell autonomous function of *Osr1* in FAPs of embryonic limbs.

None of the images does have a scale bar.

Reviewer #2 (Remarks to the Author)

Muscle connective tissue fibroblasts have previously been shown to be an important part of the niche regulating muscle progenitors during development and regeneration. These fibroblasts express several transcription factors (including *Tbx4*, *Tbx5*, and *Tcf4*) and genetic deletion of these transcription factors leads to non-cell autonomous effects on muscle development. This paper adds a new transcription factor, *Osr1*, to the list of genes expressed in fibroblasts and whose loss-of-function appears to lead to non cell-autonomous effects on muscle patterning. In addition, this paper shows that these cells can give rise to adipocytes and *Col1*⁺ fibroblasts in culture, but not bone or muscle. They also show that embryonic *Osr1*⁺ fibroblasts give rise to adult fibroblasts that express similar markers to the recently named fibroadipogenic cells (FAPs). Finally, they identify ECM components and secreted factors regulated by *Osr1*. The identification of *Osr1* as a marker of muscle connective tissue fibroblasts is a nice addition to the field and *Osr*'s functional role in fibroblasts to non cell-autonomously regulate muscle progenitors is interesting, but in need of further data. The most significant contribution of the paper is the comparison of embryonic and adult muscle connective tissue fibroblasts, but additional experiments are required in order to establish their relationship. My specific comments are below:

1. *Osr1* is identified as a marker of MCT fibroblasts in mouse via a tamoxifen inducible CreER *Osr*GCE allele. It is unclear whether *Osr1*⁺ fibroblasts are associated with all limb muscles or only

some limb muscles. This is important to establish since the *Osr1*^{-/-} mice have defects in only some limb muscles. *Osr1*^{GCE};RosaLacZ limbs stained in whole mount for b-gal activity would allow for a global view of where *Osr1* is expressed and in associated with which muscles.

2. The authors use FACS-isolated cytopun genetically labeled embryonic *Osr1* fibroblasts to compare marker expression of these fibroblasts with other known fibroblast markers (PDGFR α , Sca1, Col1, Col6, α SMA, and Tcf4). For the genetically labeled adult *Osr1* fibroblasts they compare marker expression via tissue sections. It is important that they compare marker expression on sections and in FACS isolated cells for embryonic and adult fibroblasts. While FACS-isolated cells more readily allows for quantification, antibody labeling may differ from tissue section. In the embryo, comparison of markers on sections should not be difficult as they already have sections. In the adult, FAPs have been most thoroughly identified via FACS, and therefore it is surprising that the authors have not used genetically labeled *Osr1*^{GCE} fibroblasts to determine whether these cells fall within the FAPs gate (see strategy of Joe et al. 2010). They should do this. In addition, using the FAPs FACS strategy they can then determine whether the *Osr1*-derived cells are indeed Sca1⁺. See related comment 7 below.

3. It is unclear whether *Osr1* is expressed in adult muscle connective tissue fibroblasts. It is important to give tamoxifen to adult *Osr1*^{GCE}; RosamTmG (or some other reporter) mice and determine whether *Osr1* only marks these cells in the embryo. If *Osr1* is not expressed in the adult, this does not diminish the current paper. However, *Osr1* is unlikely to be a useful marker for adult fibroblasts.

4. *Osr1* mutant analysis. The authors nicely describe the muscle anatomy defects in limbs of *Osr1*^{-/-} mice. Two questions remain unclear. First, why are only some muscles affected. Do the affected muscles correspond to regions with particularly high levels of *Osr1* expression? If not, why is only a subset of muscles affected. Second, the authors claim that the effects of *Osr1* deletion are non cell-autonomous; *Osr1*^{-/-} fibroblasts lead to mis-regulation of myogenic progenitor proliferation and survival. However, they have not formally demonstrated this. It is true that *Osr1* appears to be primarily expressed in fibroblasts, but low levels of *Osr1* in myogenic cells could cell-autonomously regulate myogenic progenitors. Ideally, the authors would conditionally delete *Osr1* in just fibroblasts or just myogenic cells (via an *Osr1*^{fl} allele). Clearly, this is beyond the scope of the paper. However, the authors could test this non-cell autonomous effect by co-culturing *Osr1*^{-/-} versus *Osr1*^{+/+} fibroblasts with myoblasts. An additional control experiment would be to isolate *Osr1*^{-/-} versus *Osr1*^{+/+} myoblasts and test whether their proliferation, differentiation, and/or survival differ in culture. If their model is correct, *Osr1*^{-/-} should be unable to support the level of myoblast proliferation and should promote higher levels of myoblast apoptosis. In addition, *Osr1*^{-/-} and *Osr1*^{+/+} myoblasts should not differ.

5. Fig. 4F. The ectopic tendon formation in *Osr1* mutants is not very convincing. A different image should be shown. Perhaps show this in whole mount?

6. Fig. 5B. Hard to see any change in Col14 structure in *Osr1* mutants.

7. The authors should be extremely careful in their use of the terms muscle connective tissue fibroblasts and FAPs. Please carefully define these at the beginning of the paper. It is my impression that FAPs are muscle connective tissue fibroblasts in the adult that have been functionally defined by FACS analysis - whereby they are CD45⁻ (not hematopoietic), CD31⁻ (not endothelial), α 7 integrin⁺ (not myogenic), and Sca1⁺. They also have been defined by their ability in vitro to make fibrogenic and adipogenic cells. Thus if you wanted to see if a similar FAPs-like population was present in the embryo you would need to do FACS analysis of embryonic muscle and show that a similar CD45⁻CD31⁻ α 7 integrin⁺ Sca1⁺ population was present. In the manuscript, the authors' data shows that genetically labeled *Osr1* embryonic cells are Sca1⁻. Does that mean these cells are not FAPs? The nice culture experiments showing *Osr1* cells can make fibrogenic and adipogenic cells in culture do support their potential dual fates. See related

comment 2 above.

9. "Osr1 controls a transcriptional program maintaining muscle connective tissue identity" subheading. The authors do show a change in gene expression in Osr1 mutants. However, they have not shown that these cells change fate in the absence of Osr1. They cite another paper showing ectopic cartilage in Osr1/2 mutants and show unconvincing data of ectopic tendon formation (see comment 5). The obvious experiment would be to repeat the in vitro assays (shown in Fig. 2A) with Osr1⁻ cells. If their model is correct Osr1⁻ cells should now make chondrocytes.

10. "Osr1⁺ developmental FAPs provide a myogenic niche in developing limbs". The authors do not show evidence that Osr1⁺ fibroblasts are supportive of myogenic cells. A co-culture experiment culturing Osr1⁻ versus Osr1⁺ with myoblasts would provide evidence of this (see comment 4). It should be noted that other previous papers have demonstrated, via fibroblast-myoblast co-cultures, that muscle connective tissue fibroblasts are supportive of myogenesis (e.g. Joe et al. 2010, Mathew et al. 2011).

Minor comments:

1. The authors should be careful of their wording. "Given the supportive function of adult FAPs..., we asked whether embryonic Osr1⁺ FAP-like cells fulfill a similar function in mouse embryos". However, their experiment does NOT test the function of the cells, rather it tests the function of Osr1 gene.

2. Scale bars are absent on most panels.

3. It would be useful if the authors present a model of their data.

4. The subheading "Osr1 function is required for embryonic myogenesis" is not factually correct. Osr1 mutant mice do develop muscle, thus Osr1 is not required. "Osr1 regulates embryonic myogenesis" is a more accurate title.

Reviewer #3 (Remarks to the Author)

A. Summary of the key results

This manuscript traces the developmental lineage of fibro-adipogenic precursor cells (FAPs) via expression of the transcription factor Odd skipped-related 1 (Osr1). It also established a role for FAPs and Osr1 in development, as deletion of Osr1 leads to defects in muscle morphogenesis. The mechanism proposed is that Osr1 is required in FAPs to secrete pro-myogenic ECM components and signaling molecules, based on gene expression analysis.

B. Originality and interest: if not novel, please give references

It is increasingly well established that FAPs modulate myogenic cell activity in the adult, and this manuscript extends those observations to development- this is novel and significant. It also provides potential mechanisms for regulation of myoblast activity by FAPs (ECM and chemokine secretion); these would be more significant were they tested, but I understand that is outside the scope of the current manuscript.

C. Data & methodology: validity of approach, quality of data, quality of presentation

The experiments presented appear to have been done and analyzed appropriately; in addition, the data presented are beautiful. In particular, the muscle whole mount preparations of Osr1^{-/-} mice in Figure 3 are striking. It may be useful to provide scale bars to aid the reader.

D. Appropriate use of statistics and treatment of uncertainties

Yes.

E. Conclusions: robustness, validity, reliability

In general the conclusions are sound, although the authors do occasionally overstate their case. For example, FAPs/Osr1 are not 'required for myogenesis'- it still occurs. They should consider toning down this and other such statements.

F. Suggested improvements: experiments, data for possible revision

Is Osr1 expression maintained in adult FAPs? It would be both useful and interesting to know.

G. References: appropriate credit to previous work?

Yes.

H. Clarity and context: lucidity of abstract/summary, appropriateness of abstract, introduction and conclusions

The manuscript is generally well-written, but I would suggest again that the authors moderate some of their conclusions.

Reviewer #1 (Remarks to the Author):

Garcia et al provide an interesting but mechanistically not very deep description of the role of the zinc-finger transcription factor Osr1 for formation of skeletal muscles during limb development of mice. The authors found that Osr1 is expressed in a subset of muscle connective tissue cells (MCT) during development sharing characteristics of so-called fibroadipogenic progenitor cells (FAPs), which is another word for mesenchymal stem cells in skeletal muscle. Inactivation of Osr1 resulted in malformation of muscles in the limbs but did not affect distribution of Osr1+ cells. RNA-seq analysis revealed a strong up-regulation of tendon and cartilage genes in Osr1-cells, which corresponds to earlier findings in neural crest-derived mesenchyme of tongue muscles (Liu et al, PNAS, 2013). In parallel, the authors detected reduced concentration of mRNAs for collagen6 (at E11.5) and collagen 3 and collagen5 (at E13.5) as well as a decrease of transcripts of several chemokines in Osr1-deficient cells. The authors conclude that the lack of Osr1 impairs muscle progenitor cell proliferation and positioning without dramatic effects on limb muscle progenitor cell migration.

The principal observation that lack of Osr1 disturbs formation of individual skeletal muscles during mouse limb development is interesting. It also seems likely that the dysregulation of ECM genes and chemokines in Osr1-mutant mesenchymal cells in the limb contribute to skeletal muscle limb formation. However, the whole analysis remains rather superficial. It does not become clear which effects in the Osr1-mutant mesenchymal cells are primary or secondary and what the specific contribution of changes in Osr1-mutant mesenchymal cells to skeletal muscle malformation is. The authors propose that Osr1 identifies a subpopulation embryonic FAPs, which is certainly true, but a more comprehensive quantitative analysis that would define the relative contribution of Osr1+ cells to FAPs and the relation of different subpopulation of FAPs to each other is missing.

Specific comments

1. The authors detected several genes that were either up- (261 genes) or downregulated (250 genes) in Osr1 mutant cells. Some of the dysregulated genes will most likely contribute to the phenotype of Osr1 mutants. However, the authors did not make any efforts to evaluate the individual contribution of any of the dysregulated genes. Is the downregulation of collagen 6 the major culprit? Is there a role for any of the other downregulated collagens? What is the impact of the downregulation of chemokines? I do not ask for a comprehensive assessment of all changes but a more in-depth functional analysis is absolutely mandatory.

The key message we aim to convey is the concept of a niche environment that is provided by Osr1+ cells and that is necessary as a whole, i.e. as a combination of diverse cues. We characterized the composition of this niche and showed that Osr1 is essential for the niche. Our RNA-Seq experiment provides evidence that this niche is composed of secreted signaling factors and a specific ECM, which in Osr1 deficient embryos shifts from a muscle-ECM composition to a cartilage / tendon ECM composition. However we agree with the reviewer that such description of a combinatorial niche lacks mechanistic depth as to the contribution of individual components it consists of. We have taken the following approaches to address this point:

We have added two in vitro models clearly demonstrating a cell-non autonomous role of Osr1+ cells via ECM production (Fig. 9c) and soluble factor signaling (Fig. 9e) on myogenesis.

We have acquired two mouse mutants that we propose as models for either an isolated ECM defect (Col6a1 null where synthesis of mature COL6 is abrogated) or a chemokine signaling defect, in this case specifically Cxcl12 (Cxcr4 null, the receptor for Cxcl12 that is expressed in myogenic progenitors, see Vasyutina et al. 2005 Genes Dev. 19). We show that both mutants exhibit overlapping defects with those of Osr1 mutants.

In Col6a1 mutants we demonstrated a significant reduction in the number of myogenic cells at E11.5 (Fig. 9d) consistent with the reduction observed in Osr1 mutants, although being weaker than in Osr1 mutants. However, no clear-cut reproducible patterning phenotype was seen in Col6a1 mutant embryos at E14.5 with our methods. This indicates that the loss of COL6 alone causes a transient reduction of the myogenic cell pool. This result highlights that in Osr1 mutants, Col6 genes are not the only affected genes, but also other genes encoding vital components of the ECM including e.g. SLRPs vital for fibrillary collagen assembly are downregulated, emphasizing a combined ECM effect. As noted above, our in vitro model clearly demonstrates the defective ECM production by Osr1-deficient cells (Fig 9c). We have in addition confirmed the downregulation of the SLRPs Decorin, Lumican and Nidogen2 via qPCR (Fig. 7b). Moreover, we now show that Col6a1 and Col6a3 are direct Osr1 target genes (Fig. 7e, f), supporting the notion that Osr1 controls global ECM formation via direct regulation of key components.

In Cxcr4 mutants, we find muscle patterning phenotypes overlapping with those of Osr1 mutants (Fig. 9f, g) supporting a role for Cxcl12 downstream of Osr1. We now also show that Cxcl12 is a direct Osr1 target (Fig. 8g). The muscle patterning defects observed in both mutants point towards a role of Osr1-Cxcl12 axis in local guidance of myogenic cells as suggested by the reviewer (see comment 6).

2. Moreover, no attempts were made to understand the role of Osr1 in the pro-myogenic transcriptional program that enables FAPs to control skeletal formation in the lab. Some genes were downregulated were other were upregulated. Therefore, it seems very likely that several transcriptional changes in Osr1 mutant cells are secondary events (e.g. via Sox9 and others). A ChiP-Seq experiment might help to identify some of the primary targets of Osr1.

We indeed observed up and downregulated genes in Osr1 defective cells. We interpret this as a shift from a muscle connective tissue signature towards a cartilage / tendon signature (Fig. 6d, e). We also now provide further evidence for an increased propensity of Osr1 deficient cells to undergo chondrogenic differentiation (Fig. 6f). However we agree with the reviewer that this falls short of distinguishing direct from indirect events. We decided to perform ChIP experiments focusing on deregulated candidate genes. Since there is no working antibody for Osr1 (we have tested three commercially available ones, all failed in ChIP), we used a low-expression lentiviral construct carrying Osr1-tripleFLAG. The approach was to search for Osr1 consensus binding sites either within upstream promoter regions, or within DNaseI hypersensitive regions distantly upstream of candidate targets (but within the same topological domain (Dixon et al. 2012 Nature 485). ChIP experiments were performed for candidate binding sites at the Col6a1, Col6a3, Lumican and Cxcl12 loci. We could show

that *Col6a1*, *Col6a3*, and *Cxcl12* are direct *Osr1* targets (Fig. 7e, f, 8f, g) supporting our hypothesis of a combined ECM / signaling factor niche.

3. The authors claim that the function of muscle connective tissue cells during embryonic development and of FAPs during adult muscle regeneration might be similar without providing a clear proof for this hypothesis. Since *Osr1* is widely expressed throughout the intermediate mesoderm and activated in many other tissues, including the branchial arches and limb buds at later embryonic stages, it does not seem to specifically define the embryonic origin of FAPs but rather identifies a specific subset of muscle connective tissue cells required for correct skeletal muscle patterning of the limb. The proposed parallel to FAPs in adult mice is superficial and poorly substantiated, in particular since the authors did not even take a superficial glimpse on adult mice. Formally, it is possible that *Osr1*⁺ cells only generate a transient subpopulation of FAPs during limb muscle formation.

The reviewer is very likely referring to the first two sentences in the abstract; we aimed to point out functional similarities that clearly exist between embryonal MCT and adult FAPs, namely a myogenesis-supportive role. We rephrased the first two sentences of the abstract: "Fibro-adipogenic progenitors (FAPs) are an interstitial cell population in adult muscle, which is part of the muscle connective tissue (MCT) and supports muscle homeostasis and regeneration of muscle. During development, MCT cells provide a supportive function where they control muscle patterning."

*The roles of embryonal MCT and adult FAPs are, as we point out, mechanistically not well clarified yet, especially in adult FAPs. Our study provides mechanistic insights employed by embryonal FAPs, which we propose as a regional subpopulation of embryonal MCT in the limb. This point was further analyzed in revision, as we now performed a comparative analysis of *Osr1* expression with the known MCT marker *Tcf4* (Fig. 2; see also comment 5)*

*We have also added data analyzing postnatal *Osr1* expression and show that *Osr1* is downregulated after birth and not expressed in adult FAPs at a level suitable for e.g. identification on tissue sections (Supplementary Fig. 6b, c). However this does not denote that *Osr1*⁺ cells are a transient population, since we have also added substantial additional data into the lineage analysis of *Osr1*⁺ developmental cells and are now able to clearly show that *Osr1*⁺ limb cells are predecessors of adult FAPs, but no other muscle interstitial population (Fig. 4c, d). FACS data were confirmed by cell counting on sections (Fig. 4e), this analysis also confirms a regionalized level of *Osr1*⁺ cell contribution to FAPs correlating with the embryonal expression pattern (see comment 4).*

*Importantly we also provide FACS data from E18.5 embryos that for the first time shows the emergence of a *Sca1*-expressing population in developing muscle that overlaps with the *Osr1*⁺ population (Supplementary Fig. 6a). This further supports the lineage continuum between *Osr1*⁺ developmental cells and adult FAPs.*

4. It is not clear to what degree *Osr1*⁺ cells contribute to the FAP/MSK population in different adult muscles. Do *Osr1*⁺ cells contribute to FAPs in other adult muscles? (If they really contribute to a major degree to adult FAPs). Is only the patterning of limb muscles affected in *Osr1* mutants?

We focused our analysis on limb muscles since this is the classical model of cell-non autonomous instruction of invading myogenic cells by the local environment. We have added data showing

expression of Osr1 in interstitial cells in other muscles in the embryo (shoulder girdle, back, tongue, diaphragm; Supplementary Fig. 1b, c). Defects in e.g. shoulder girdle muscles are pointed out in new Supplementary Fig. 7f.

We strongly extended our analysis of Osr1 descendant contribution to FAPs. FACS analysis shows clearly that Osr1 descendants in muscle connective tissue exclusively contribute to the FAP pool (Fig. 4c, d). Moreover we have performed section immunolabelling analysis and quantification of Osr1 cell descendant contribution to FAPs in two different limb muscles, representing a muscle with high Osr1 expression during development (m. gastrocnemius) and a muscle with lower Osr1 expression in development (Tibialis anterior). This analysis shows that differential contribution to the FAP pool can be observed correlating with the degree of developmental Osr1 expression. In addition we have analyzed the diaphragm as a non-limb muscle and show that Osr1+ embryonal cells substantially contribute to FAPs in this muscle as well (Fig. 4e).

5. The authors convincingly identify Osr1 as a marker of a subpopulation of MCT cells in the embryonic limb and found a partial overlap with Tcf4 expression, a popular marker of MCT cells in the developing limb. They also analyzed the expression of several typical markers of mesenchymal stem cells such PDGFRalpha. However, the relative input of Osr1+ cells to MCT in the limb remains enigmatic. It should be relatively easy to determine the overall contribution of Osr1+ cells to PDGFRalpha cells (FACS, co-staining on sections) in comparison to Tcf4 and to other markers. The co-staining of Osr1 with Tcf4 needs to be done in both directions to see whether all Tcf4 cells express Osr1 or not. The staining for Tcf4 on FACS-sorted Osr1 cells does not provide this information. Are Osr1+/Tcf4+, Osr1+/Tcf4- and Osr1-/Tcf4+ cells homogenously distributed in the muscle anlagen of the developing limb? Such approaches would help to substantiate the hypothesis that a combination of instructive transcription factors in the MCT determines the local character of individual muscle anlagen.

We have in depth analyzed the relationship of Osr1 and Tcf4. First, we have compared the expression patterns of Osr1 and Tcf4 on E11.5 and E13.5 hindlimb sections (Fig. 2a, b). This already clearly shows that Osr1 and Tcf4 label distinct MCT population but partially overlap, which is consistent with the cytospin data (Fig. 1h). We provide examples of muscles with high Osr1 and low Tcf4 expression and vice versa. We also show colabelling for Osr1(GFP) and PDGFRa on E13.5 hindlimb sections (Supplementary Fig. 5b) showing a widespread overlap of expression in line with the cytospin (Fig. 1h) and FACS (Supplementary Fig. 4a) data.

We additionally analyzed a second developmental time point, E18.5. we extracted GFP+ and GFP-muscle interstitial cells (lin-,a7-integrin-) by FACS, thus comprising all non-myogenic interstitial cells. We show that Osr1 and Tcf4 show a much higher degree of coexpression at that stage (Fig. 2c) compared to E13.5.

Moreover we extended the lineage tracing experiments and show that Osr1+ progenitors from E11.5 or p0 become Tcf4+ by E18.5 (Fig. 3c) and in adult (Fig. 4e), respectively. This indicates that Osr1+ early MCT cells activate Tcf4 expression later. This is in line with data from Mathew et al. 2011 (Development 138) indicating that in neonatal mice MCT of virtually all limb muscles is positive for Tcf4. We also show that in parallel to MCT cells assuming Tcf4 expression, Osr1 expression is lost (Supplementary Fig. 6), so that Osr1 is not a marker for adult MCT or FAPs.

6. The authors noted reduced presence and reduced proliferation of muscle progenitor cells in Osr1 mutant, which lead to the hypothesis that OSR1+ FAPs directly stimulate proliferation of muscle progenitor cells. The authors hypothesize that the reduced expression of chemokines by Osr1 mutant FAPs might be responsible for this phenomenon. However, chemokines have been mostly associated with migration of muscle progenitor cells. The effects on proliferation are usually rather weak. It also seems possible that the aberrant distribution of muscle progenitor cells in Osr1 mutant limbs or a disturbed ECM composition contributes to reduced proliferation.

Thank you for pointing this out, we agree with this notion. We have extended this analysis and specified the conclusions. We have now analyzed Cxcr4 mutant mice and show a muscle patterning phenotype strikingly overlapping with that observed in Osr1 mutants, specifically in the biceps femoris muscles, where a lack of myogenic progenitors is seen at the sites these muscles should form compatible with a defect in local migration of these cells to their normal target areas.

We have also analyzed Col6a1 mutant mice that show a similar (but milder) reduction in the myogenic cell pool there, which is not the case for Cxcr4 mutants (Vasyutina et al. 2007 Genes Dev 19). Moreover we have performed an in vitro ECM assay showing that the ECM produced by Osr1 null cells is incapable of promoting myogenic cell proliferation. This leads to a hypothesis as suggested by the reviewer, that ECM deposition is required for cell survival and proliferation, while chemokine signaling appears to mainly influence the local migration / distribution of myogenic cells.

7. Due to the embryonic lethality of germ line Osr1 mutants it is difficult to explore long-term effects and to verify the loss of individual muscles in newborn mice. Obviously, it would be much better to inactivate the Osr1 gene with PDGFRalpha-Cre mice, although such an experiment requires substantial efforts and takes a long time. Nevertheless, such an approach would also validate a cell autonomous function of Osr1 in FAPs of embryonic limbs.

We appreciate that a conditional inactivation of Osr1 with a Cre driver that allows for a follow-up analysis would be helpful. However, we think that PDGFRa may not be a good choice in this context. While our population of embryonal FAPs is to almost 100% PDGFRa positive, in the embryo PDGFRa is also a general mesenchymal marker expressed in many other tissues. Very likely this strategy would not circumvent embryonic lethality, since Pdgfra is also expressed in the heart atria, valves and aortic outflow tract:

http://www.eurexpress.org/ee/databases/assay.jsp?assayID=euxassay_010504&image=01

We have now in addition performed two independent in vitro assays showing a non-cell autonomous function of Osr1+ cells via ECM (Fig. 9c) and secreted factors (Fig. 9e).

8. None of the images does have a scale bar.

We have added scale bars

Reviewer #2 (Remarks to the Author):

Muscle connective tissue fibroblasts have previously been shown to be an important part of the niche regulating muscle progenitors during development and regeneration. These fibroblasts express several transcription factors (including Tbx4, Tbx5, and Tcf4) and genetic deletion of these transcription factors leads to non-cell autonomous effects on muscle development. This paper adds a new transcription factor, Osr1, to the list of genes expressed in fibroblasts and whose loss-of-function appears to lead to non cell-autonomous effects on muscle patterning. In addition, this paper shows that these cells can give rise to adipocytes and Col1+ fibroblasts in culture, but not bone or muscle. They also show that embryonic Osr1+ fibroblasts give rise to adult fibroblasts that express similar markers to the recently named fibroadipogenic cells (FAPs). Finally, they identify ECM components and secreted factors regulated by Osr1. The identification of Osr1 as a marker of muscle connective tissue fibroblasts is a nice addition to the field and Osr's functional role in fibroblasts to non cell-autonomously regulate muscle progenitors is interesting, but in need of further data. The most significant contribution of the paper is the comparison of embryonic and adult muscle connective tissue fibroblasts, but additional experiments are required in order to establish their relationship. My specific comments are below:

1. Osr1 is identified as a marker of MCT fibroblasts in mouse via a tamoxifen inducible CreER OsrGCE allele. It is unclear whether Osr1+ fibroblasts are associated with all limb muscles or only some limb muscles. This is important to establish since the Osr1-/- mice have defects in only some limb muscles. OsrGCE;RosaLacZ limbs stained in whole mount for b-gal activity would allow for a global view of where Osr1 is expressed and in associated with which muscles.

Thank you for this notion. The experiment suggested by the reviewer did not deliver the expected results, as Osr1 expression in the dermis led to an overall diffuse staining. We have addressed this issue by extending the expression analysis of Osr1 on tissue sections. We have added a new Figure, Fig. 2 that shows Osr1 expression compared to the established MCT marker Tcf4. We denote muscles with high and low Osr1 expression (biceps femoris and gastrocnemius as examples of high expression, extensor digitorum longus, tibialis anterior and peroneus as examples of low expression). This correlates with the phenotype observed in these muscles (biceps femoris in Fig. 5a, gastrocnemius, tibialis anterior, extensor digitorum longus and peroneus (the two latter were newly added) in Supplementary Fig 7d,e).

Furthermore, we also performed comparative lineage analysis on sections using the gastrocnemius and tibialis anterior muscles as examples for Osr1-high and Osr1-low muscles, respectively. This provides a contribution of Osr1 lineage cells to the PDGFRalpha (FAP) pool, which correlates with the embryonic Osr1 expression in these muscles (Fig. 3c, 4e).

2. The authors use FACS-isolated cytopun genetically labeled embryonic Osr1 fibroblasts to compare marker expression of these fibroblasts with other known fibroblast markers (PDGFRa, Sca1, Col1, Col6, aSMA, and Tcf4). For the genetically labeled adult Osr1 fibroblasts they compare marker expression via tissue sections. It is important that they compare marker expression on sections and in FACS isolated cells for embryonic and adult fibroblasts. While FACS-isolated cells more readily allows for quantification, antibody labeling may differ from tissue section. In the

embryo, comparison of markers on sections should not be difficult as they already have sections. In the adult, FAPs have been most thoroughly identified via FACS, and therefore it is surprising that the authors have not used genetically labeled Osr1GCE fibroblasts to determine whether these cells fall within the FAPs gate (see strategy of Joe et al. 2010). They should do this. In addition, using the FAPs FACS strategy they can then determine whether the Osr1-derived cells are indeed Sca1+. See related comment 7 below.

Embryo: we have now compared Osr1 and Tcf4 expression on tissue sections at E11.5 and E13.5 (Fig. 2a, b) showing the low partial overlap of both markers. We also compared Osr1 and PDGFRalpha expression at E13.5 on tissue sections and found a widespread overlap (Supplementary Fig. 5b), which is fully consistent with the FACS and cytospin data (Figure 1a, Supplementary Fig. 5a).

We aimed to extend this analysis to E18.5, however at this stage Osr1-GCE expression apparently has decreased in muscle interstitial cells and we were not satisfied with the quality of labelling on sections. Osr1-GFP expression was, however, still sufficient to isolate Osr1-GFP+ and Osr1-GFP- populations from E18.5 muscles by FACS. Of note, the combination of GFP+ and GFP- cells provides the “full picture” of muscle interstitial cells (lin-;a7-integrin-) at that stage. We compared Tcf4 and PDGFRalpha expression in these populations (Fig. 2c) and found that Osr1, Tcf4 and PDGFRalpha largely overlap.

Adult: we provide data showing that Osr1 expression fades during development (especially during the first three postnatal weeks) and is barely detectable even by FACS in adult FAPs (FACS strategy according to Joe et al., Sca1 as FAP marker). For the lineage analysis, we have performed FACS analysis as suggested. Osr1GCE;mTmG animals were Tam pulsed at p0/p1. FAPs (Joe et al. Protocol) were isolated from adult mice. We also isolated Sca1- cells and satellite cells. We clearly show that Osr1 progeny exclusively contributes to Sca1+/CD34+ FAPs (Fig. 4c). In addition, cytospin experiments performed with the Sca1+/CD34+/mG+ cells show that Osr1 lineage cells are mostly positive for PDGFRalpha, a recognized marker for FAPs (Uezumi et al. Nat Cell Biol 2010) (Fig. 4d). In addition we show that Osr1 lineage FAPs mostly express Tcf4, and that indeed the majority of adult FAPs (irrespective of Osr1 lineage) express Tcf4 (Fig. 4d) as previously suggested by Matthew et al. (2011).

3. It is unclear whether Osr1 is expressed in adult muscle connective tissue fibroblasts. It is important to give tamoxifen to adult Osr1GCE; RosamTmG (or some other reporter) mice and determine whether Osr1 only marks these cells in the embryo. If Osr1 is not expressed in the adult, this does not diminish the current paper. However, Osr1 is unlikely to be a useful marker for adult fibroblasts.

We now show data indicating a decrease of Osr1 expression soon after birth (Supplementary Fig. 6c). We have attempted to detect Osr1-GFP on adult mouse muscle sections, but could not detect expression. Moreover, FACS analysis of adult FAPs (CD54-/Ter119-/CD31-/a7-integrin-/Sca1+) detected borderline GFP expression in FAPs (Supplementary Fig. 6b). Based on this low level of expression, we conclude that employing the Osr1-driven Cre recombinase in adult mice would be unsuccessful.

4. Osr1 mutant analysis. The authors nicely describe the muscle anatomy defects in limbs of Osr1^{-/-} mice. Two questions remain unclear. First, why are only some muscles affected. Do the affected muscles correspond to regions with particularly high levels of Osr1 expression? If not, why is only a subset of muscles affected. Second, the authors claim that the effects of Osr1 deletion are non cell-autonomous; Osr1^{-/-} fibroblasts lead to mis-regulation of myogenic progenitor proliferation and survival. However, they have not formally demonstrated this. It is true that Osr1 appears to be primarily expressed in fibroblasts, but low levels of Osr1 in myogenic cells could cell-autonomously regulate myogenic progenitors. Ideally, the authors would conditionally delete Osr1 in just fibroblasts or just myogenic cells (via an Osr1^{fl} allele). Clearly, this is beyond the scope of the paper. However, the authors could test this non-cell autonomous effect by co-culturing Osr1^{-/-} versus Osr1^{+/+} fibroblasts with myoblasts. An additional control experiment would be to isolate Osr1^{-/-} versus Osr1^{+/+} myoblasts and test whether their proliferation, differentiation, and/or survival differ in culture. If their model is correct, Osr1^{-/-} should be unable to support the level of myoblast proliferation and should promote higher levels of myoblast apoptosis. In addition, Osr1^{-/-} and Osr1^{+/+} myoblasts should not differ.

See comment 1; we have comparatively analyzed Osr1 expression in several limb muscles and found that the muscle phenotype is correlated with regions of high Osr1 expression in MCT.

We have addressed the non-cell autonomous effect of Osr1⁺ cells in two ways. First, we added a second line of evidence suggesting that Osr1 is not expressed, even at low level, in myogenic cells. We cytopun GFP⁺ and GFP⁻ FACS isolated E13.5 limb cells (i.e. in combination representing all limb mesenchymal mononuclear cells) and stained for Myf5 and Myogenin, in addition to antibody staining for GFP to detect even low expression. No myogenic cells were detected in the GFP⁺ pool. Coexpression of GFP and Myf5 or Myogenin was never observed (Supplementary Fig. 3). Second, we performed two complementary in vitro assays that demonstrate non-cell autonomous effects of Osr1 cells (Fig. 9c, e; see comment 10).

5. Fig. 4F. The ectopic tendon formation in Osr1 mutants is not very convincing. A different image should be shown. Perhaps show this in whole mount?

As we noted in the text, we detected ectopic expression of Scx via in-situ hybridization (which is an order of magnitude less sensitive than RT-qPCR) solely at the anatomical location shown in figure 4f of the original manuscript. This was reproducible, however the picture shown is the best we have available. We think that it shows ectopic Scx expression, but we agree that it is a subtle alteration. As suggested, we have performed whole-mount ISH for Scx, which supports Scx upregulation and aberrant expression (Supplementary Fig. 10b). We have moved the panel to the supplementary data (Supplementary Fig. 10a) and have instead inserted the chondrogenic assay data (see comment 9) at this position (now Fig. 6f).

6. Fig. 5B. Hard to see any change in Col14 structure in Osr1 mutants.

The panel (now Fig. 9b) clearly shows altered Col14 expression in muscle interstitium.

7. The authors should be extremely careful in their use of the terms muscle connective tissue fibroblasts and FAPs. Please carefully define these at the beginning of the paper. It is my

impression that FAPs are muscle connective tissue fibroblasts in the adult that have been functionally defined by FACS analysis - whereby they are CD45- (not hematopoietic), CD31- (not endothelial), alpha7 integrin 7- (not myogenic), and Sca1+. They also have been defined by their ability in vitro to make fibrogenic and adipogenic cells. Thus if you wanted to see if a similar FAPs-like population was present in the embryo you would need to do FACS analysis of embryonic muscle and show that a similar CD45-CD31-alpha7 integrin- Sca1+ population was present. In the manuscript, the authors' data shows that genetically labeled Osr1 embryonic cells are Sca1-. Does that mean these cells are not FAPs? The nice culture experiments showing Osr1 cells can make fibrogenic and adipogenic cells in culture do support their potential dual fates. See related comment 2 above.

Thank you for pointing this out. We have attempted to clarify this in the text.

FACS analysis of embryonic muscle: Sca1 is expressed in the embryo in the tail and the aorta-gonad-mesonephros region at E11.5 (Miles et al. 1997 Development); other developmental time points have so far not been investigated. In adult mouse muscle Sca1 marks FAPs. This led us to speculate that FAPs, which arise from Osr1+/Sca1- progenitors, must acquire Sca1 expression later. We now show data substantiating this hypothesis. We performed FACS analysis of E18.5 muscles as suggested by the reviewer (CD54-/Ter119-, CD31-, a7-integrin-) and analyzed Osr1-GFP and Sca1 expression. This showed that at this time point a Sca1+ population emerged (Supplementary Fig. 6a). Intriguingly, Osr1-GFP expression was partitioned between the Sca1+ and Sca1- fractions, supporting that Sca1- progenitors during development convert to Sca1+ cells.

9. "Osr1 controls a transcriptional program maintaining muscle connective tissue identity" subheading. The authors do show a change in gene expression in Osr1 mutants. However, they have not shown that these cells change fate in the absence of Osr1. They cite another paper showing ectopic cartilage in Osr1/2 mutants and show unconvincing data of ectopic tendon formation (see comment 5). The obvious experiment would be to repeat the in vitro assays (shown in Fig. 2A) with Osr1- cells. If their model is correct Osr1- cells should now make chondrocytes.

Thank you for this suggestion. We have performed the chondrogenesis assay with Osr1-/- cells in comparison to Osr1+/-, indeed Osr1 deficient cells have a higher chondrogenic potential (Fig. 6f).

10. "Osr1+ developmental FAPs provide a myogenic niche in developing limbs". The authors do not show evidence that Osr1+ fibroblasts are supportive of myogenic cells. A co-culture experiment culturing Osr1- versus Osr1+ with myoblasts would provide evidence of this (see comment 4). It should be noted that other previous papers have demonstrated, via fibroblast-myoblast co-cultures, that muscle connective tissue fibroblasts are supportive of myogenesis (e.g. Joe et al. 2010, Mathew et al. 2011).

Thank you for this suggestion. As we propose a dual role of Osr1+ cells via ECM deposition and secreted factors, we performed two complementary experiments. First, an ECM deposition assay was performed. FACS-isolated Osr1+/- and Osr1-/- cells were plated and allowed to produce an ECM for 10 days. Then the ECM was decellularized and myoblasts were seeded on the ECM. This showed that

an ECM produced by Osr1+/-, but not by Osr1-/- cells, promoted myogenic cell proliferation (Fig. 9c). Second, we performed a transwell assay as Mathew et al. 2011. This confirmed that Osr1+/- fibroblasts promote myogenesis, while Osr1-/- cells fail to do so (Fig. 9e).

Minor comments:

1. The authors should be careful of their wording. "Given the supportive function of adult FAPs..., we asked whether embryonic Osr1+ FAP-like cells fulfill a similar function in mouse embryos". However, their experiment does NOT test the function of the cells, rather it tests the function of Osr1 gene.

We agree with this notion; we have changed the sentence to: "Given the supportive function of adult FAPs in regenerative myogenesis and the non-cell autonomous function ascribed to limb MCT cells during development, we asked whether Osr1 could mediate such a supportive function in embryonal FAP-like cells."

2. Scale bars are absent on most panels.

We have included scale bars

3. It would be useful if the authors present a model of their data.

Thank you for the suggestion. We have added a model in Fig. 9j.

4. The subheading "Osr1 function is required for embryonic myogenesis" is not factually correct. Osr1 mutant mice do develop muscle, thus Osr1 is not required. "Osr1 regulates embryonic myogenesis" is a more accurate title.

Thank you for this comment, we have changed the subheading accordingly

Reviewer #3 (Remarks to the Author):

A. Summary of the key results

This manuscript traces the developmental lineage of fibro-adipogenic precursor cells (FAPs) via expression of the transcription factor Odd skipped-related 1 (Osr1). It also established a role for FAPs and Osr1 in development, as deletion of Osr1 leads to defects in muscle morphogenesis. The mechanism proposed is that Osr1 is required in FAPs to secrete pro-myogenic ECM components and signaling molecules, based on gene expression analysis.

B. Originality and interest: if not novel, please give references

It is increasingly well established that FAPs modulate myogenic cell activity in the adult, and this manuscript extends those observations to development- this is novel and significant. It also provides potential mechanisms for regulation of myoblast activity by FAPs (ECM and chemokine secretion); these would be more significant were they tested, but I understand that is outside the scope of the current manuscript.

We have in revision deepened our analysis towards the mechanisms employed by Osr1+ cells. We show direct transcriptional targets for Osr1 (Fig. 7e, f, 8g). We also analyzed two mouse models for an ECM defect (Col6a1 null) and a chemokine signaling defect (Cxcr4 null) and show in part overlapping phenotypes (Fig. 9d, f, g). Moreover we performed two complementary in vitro assays to clearly show a non-cell autonomous effect on Osr1+ cells on myogenesis via ECM (Fig. 9c) and secreted factors (Fig. 9e).

C. Data & methodology: validity of approach, quality of data, quality of presentation

The experiments presented appear to have been done and analyzed appropriately; in addition, the data presented are beautiful. In particular, the muscle whole mount preparations of Osr1-/- mice in Figure 3 are striking. It may be useful to provide scale bars to aid the reader.

Thank you very much for the kind appreciation of our data. We have added scale bars.

D. Appropriate use of statistics and treatment of uncertainties

Yes.

E. Conclusions: robustness, validity, reliability

In general the conclusions are sound, although the authors do occasionally overstate their case. For example, FAPs/Osr1 are not 'required for myogenesis'- it still occurs. They should consider toning down this and other such statements.

Thank you very much for this suggestions. We have changed this heading to "Osr1 regulates embryonic myogenesis".

F. Suggested improvements: experiments, data for possible revision

Is Osr1 expression maintained in adult FAPs? It would be both useful and interesting to know.

We have tested this; Osr1-GFP is expressed at very low level in adult FAPs (Supplementary Fig. 6b), barely detectable by FACS. An attempt to detect Osr1-GFP expression on adult tissue sections failed. Moreover, we show that Osr1 expression decreases soon after birth (Supplementary Fig. 6c).

G. References: appropriate credit to previous work?

Yes.

H. Clarity and context: lucidity of abstract/summary, appropriateness of abstract, introduction and conclusions

The manuscript is generally well-written, but I would suggest again that the authors moderate some of their conclusions.

Reviewers' Comments:

Reviewer #1:

Remarks to the Author:

Garcia et al have submitted a revised version of the manuscript "Odd skipped-related 1 (Osr1) identifies embryonic fibro adipogenic progenitors (FAPs) and regulates a pro myogenic transcriptional program during limb development", which they describe Osr1-dependent functions in muscle connective tissue cells (MCT cells) during embryonic limb development that affect muscle formation via the extracellular matrix components and secreted signaling factors.

The authors have made massive efforts to improve the quality of the manuscript and to obtain a more in-depth analysis of the function of Osr1 for MCT cells and limb muscle formation. In particular, the authors have established an in vitro model to analyze the role of Osr1+ cells for ECM production and provision of signaling factors. Furthermore, they analyzed mouse mutants for Col6a1 and Cxcl12, which are both putative downstream targets of Osr1. The authors found that inactivation of Col6a1 and Cxcl12 partially recapitulated the Osr1-phenotype, which provides strong support to the hypothesis that multiple downstream effectors of Osr1 in MCT cells provide permissive and instructive signals for myogenic progenitor cells. This new set of experiments includes evidence that Osr1 directly binds to regulatory regions of the Col6a1 and Cxcl12 genes.

In addition, the authors further analyzed transcriptional changes in Osr1-mutant MCT1 cells, which provide evidence for a shift towards chondrogenic differentiation, clearly a highly interesting finding. Further cell lineage tracing data demonstrated that embryonic MCT cells are indeed specific progenitor cells for subpopulations of FAPs in limb muscles. This is an important new result, which clearly strengthens the manuscript. The authors propose that the decrease of Osr1 expression at later developmental stages goes along with up-regulation of Sca1 in the same cell lineage. Surprisingly, the authors did not use their Osr1-lineage tracing tool to validate this hypothesis but only rely on the comparison of Osr1+ and Sca1+ cells. To pinpoint the exact contribution of Osr1+ cells to Sca1+-cells, it would be nice to employ the same long-term lineage tracing approach used for analysis of the contribution of Osr1+-cells for PDGFRalpha and Tcf4 expressing cells, although I would not consider such an experiment as mandatory.

The original version of the manuscript lacked a careful quantification of the contribution of Osr1+ cells to PDGFRalpha and Tcf4. The authors have convincingly addressed this shortcoming in the revision and now show a detailed description of the overlap between Osr1, PDGFRalpha and Tcf4 expression in MCT cells.

Finally, the authors analyzed the impact of ECM molecules and soluble signaling factors for proliferation and migration/distribution of myogenic cells, respectively. Interestingly, they found that ECM components primarily affected proliferation of myogenic cells whereas secreted soluble factors mainly influenced local migration.

Taken together, the authors have obtained a large set of new, very interesting data, which convincingly address most of the major concerns that I initially raised.

Minor point: The authors should use a more specific nomenclature when describing Osr1+ cells. Sometimes they only refer to Osr1-expressing cells as GFP+ cells, which makes the reader wonder whether Osr1+-expressing cells or traced cells derived from Osr1+ cells are meant (both are GFP-positive although the GFP is membrane bound in case of Cre-dependent reporter gene activation). This can be easily fixed.

Reviewer #2:

Remarks to the Author:

The authors have done an excellent job answering my previous comments. In particular, I am impressed with the careful comparative analysis of Osr1+ cells with other markers of muscle connective tissue fibroblasts - Pdgfra, Tcf4, and FAPs (as identified by Joe et al. FACS strategy). I also think the Osr1 +/- co-culture with myogenic cells and Osr1 +/- matrix with myogenic cells are an excellent addition to the paper. Overall, I think this paper is a strong contribution to the field and will be much appreciated.

I do have a few comments to improve the manuscript:

1. Line 35 Abstract: "developmental limb MCT cells expressing the transcription factor Osr1 give rise to adult FAPs". Since their lineage data shows that Osr1 lineage gives rise to a small subset of adult FAPs, I think this sentence should be modified to "developmental limb MCT cells expressing the transcription factor Osr1 give rise to a subset of adult FAPs".

2. Line 42 Abstract: "We conclude that embryonic FAPs provide a pro-myogenic niche required for myogenesis in developing limbs..." That have not demonstrated that Osr1+ embryonic FAPs are required. Loss of Osr1 has relatively minor effects on muscle patterning and they have not done the experiment deleting Osr1+ cells (e.g. using Osr1GCE; RosaDTA mice). It would be more accurate to state "We conclude that Osr1+ embryonic FAPs provide a pro-myogenic niche that regulates myogenesis in developing limbs..."

3. Line 24 Introduction: "muscle stem cells (satellite cells)" change to "muscle stem cells (termed satellite cells in the adult)". Satellite cells are a term reserved for adult (older than 6 weeks) muscle stem cells which reside under the basement membrane of myofibers. The term "satellite cells" is not used for embryonic, fetal, or neonatal muscle stem cells.

4. Line 54 Introduction: "It is widely accepted that myogenic cells do not contain intrinsic information that govern their place and time of differentiation" references
Ordahl and Dourain 1992
Aoyama and Asamoto 1988
Lance-jones Jones DB 408-419 1988
Lance-jones Jones DB 394-407 1988

This sentence is only true for vertebrate muscle. Please change sentence to read "It is widely accepted that vertebrate myogenic cells do not contain intrinsic information that govern their place and time of differentiation". In fact, fly muscle is intrinsically specified.

Also, not all the references are the appropriate references to cite.

Please use

Ordahl and Dourain 1992
Lance-jones Jones DB 394-407 1988 (only this Lance-Jones reference is appropriate)

5. Line 69-70 Introduction: "The pro-myogenic function ascribed to FAPs during adult muscle regeneration is an intriguing parallel to the function lateral plate-derived MCT progenitors play during embryonic myogenesis. Whether or not these two cell types are related and what molecular mechanisms contribute to their pro-myogenic function is mostly unknown." The relationship between adult and embryonic muscle connective tissue fibroblasts has been previously described in Mathew et al. 2011. At the time of the Mathew publication, FAPs had just been described and so this paper did not explicitly compare embryonic fibroblasts with adult FAPs. However, the paper clearly described (as shown by Tcf4 lineage experiments) that the lateral plate is the source of embryonic and adult muscle connective tissue fibroblasts. This paper should be cited here and in the Discussion in the paragraph beginning on Line 475.

6. Line 182-184 Results: "Intriguingly, GFP+ cells were equally distributed between Sca1⁻ and Sca1⁺ MCT cells (Supplementary Fig. 6a). This suggests that Osr1⁺ cells gain Sca1 expression over time". A similar conclusion is stated in Lines 47-478 Discussion: "An interesting notion is that adult Sca1⁺ FAPs originate from developmental Osr1⁺/Sca1⁻ cells". I am unclear whether the authors have the data to support the statement that Osr1⁺Sca1⁻ embryonic cells give rise to Sca1⁺ FAPs. The data needed would be to analyze adult Osr1⁺GCE/+;RosatdTom mice given tam at E11.5 (when Osr1⁺ cells are confirmed to be Sca1⁻) via FACS and determine whether Tom⁺ cells ended up in FAPs gate and so are Sca1⁺. The Osr1⁺ Sca1⁺ cells in adults could be due to the expansion of a small de novo population of Osr1⁺Sca1⁺ cells. As far as I can tell from the data presented, they do not have any lineage data to support that Osr1⁺Sca1⁻ cells subsequently become Osr1⁺Sca1⁺. They do not necessarily have to do this lineage experiment, but if they do not have such data they should be careful in their interpretation and conclusions.

7. Line 205-207 Results: "Osr1 progeny was observed exclusively in FAPs, while no lineage progeny was found in satellite cells or osteo-chondrogenic progenitors (Fig. 4c) showing that Osr1⁺ progenitors only contribute to the FAP pool." It is important that they explicitly mention that only 7% of the FAPs were derived from Osr1⁺ cells genetically labeled at P0. Thus Osr1⁺ cells are unlikely to be the only developmental source of FAPs.

8. Line 326 Results: "...we conclude Osr1 is a key transcription factor that maintains embryonic FAP cell identity." Given that Osr1 is not expressed in all embryonic muscle connective tissue fibroblasts, this statement only applies to the subset of fibroblasts that express Osr1. Please modify this sentence.

9. Lines 372-375 Results: "ECM produced by Osr1⁺GCE/+ cells significantly enhanced myogenic cell proliferation in comparison to cultivation on Matrigel. ECM produced by Osr1 deficient cells (Osr1⁻GCE/GCE) failed to promote myoblast proliferation in comparison to Matrigel (Fig. 9c)." Fig 9c shows that Ki67⁺Pax7⁺/Pax7⁺ cells when cultured on ECM produced by Osr1⁺GCE/+ and Osr1⁻GCE/GCE. However, the Fig does not compare these numbers when cultured just on Matrigel.

10. The title should be modified: "Odd skipped-related 1 (Osr1) identifies a population of embryonic fibro-adipogenic progenitors (FAPs) and regulates a pro-myogenic transcriptional program during limb development".

Responses to Reviewers

We are very grateful to the reviewers for the critical assessment of our work and the numerous insightful comments that helped us to improve our manuscript. We are also grateful for the reviewers' acknowledgement for the efforts performed during revision. Please find our point-by-point responses to your questions and suggestions below.

Reviewer #1 (Remarks to the Author):

Garcia et al have submitted a revised version of the manuscript "Odd skipped-related 1 (Osr1) identifies embryonic fibro adipogenic progenitors (FAPs) and regulates a pro myogenic transcriptional program during limb development", which they describe Osr1-dependent functions in muscle connective tissue cells (MCT cells) during embryonic limb development that affect muscle formation via the extracellular matrix components and secreted signaling factors.

The authors have made massive efforts to improve the quality of the manuscript and to obtain a more in-depth analysis of the function of Osr1 for MCT cells and limb muscle formation. In particular, the authors have established an in vitro model to analyze the role of Osr1+ cells for ECM production and provision of signaling factors. Furthermore, they analyzed mouse mutants for Col6a1 and Cxcl12, which are both putative downstream targets of Osr1. The authors found that inactivation of Col6a1 and Cxcl12 partially recapitulated the Osr1-phenotype, which provides strong support to the hypothesis that multiple downstream effectors of Osr1 in MCT cells provide permissive and instructive signals for myogenic progenitor cells. This new set of experiments includes evidence that Osr1 directly binds to regulatory regions of the Col6a1 and Cxcl12 genes.

In addition, the authors further analyzed transcriptional changes in Osr1-mutant MCT1 cells, which provide evidence for a shift towards chondrogenic differentiation, clearly a highly interesting finding. Further cell lineage tracing data demonstrated that embryonic MCT cells are indeed specific progenitor cells for subpopulations of FAPs in limb muscles. This is an important new result, which clearly strengthens the manuscript. The authors propose that the decrease of Osr1 expression at later developmental stages goes along with up-regulation of Sca1 in the same cell lineage. Surprisingly, the authors did not use their Osr1-lineage tracing tool to validate this hypothesis but only rely on the comparison of Osr1+ and Sca1+ cells. To pinpoint the exact contribution of Osr1+ cells to Sca1+-cells, it would be nice to employ the same long-term lineage tracing approach used for analysis of the contribution of Osr1+-cells for PDGFRalpha and Tcf4 expressing cells, although I would not consider such an experiment as mandatory.

REPLY: We're not entirely sure which experiment the reviewer is referring to. We performed long-term lineage tracing and quantified the contribution of Osr1 progeny to adult Sca1+ cells (Fig. 4c). If the reviewer is referring to the question, whether early (E11.5 or E13.5) Osr1+ cells contribute to the emerging Sca1+ cells at E18.5, indeed this experiment was not performed. Referring to this please see also the reply to Reviewer 2 comment 6.

The original version of the manuscript lacked a careful quantification of the contribution of Osr1+ cells to PDGFRalpha and Tcf4. The authors have convincingly addressed this shortcoming in the revision and now show a detailed description of the overlap between Osr1, PDGFRalpha and Tcf4 expression in MCT cells.

Finally, the authors analyzed the impact of ECM molecules and soluble signaling factors for proliferation and migration/distribution of myogenic cells, respectively. Interestingly, they found that ECM components primarily affected proliferation of myogenic cells whereas secreted soluble factors mainly influenced local migration.

Taken together, the authors have obtained a large set of new, very interesting data, which convincingly address most of the major concerns that I initially raised.

Minor point: The authors should use a more specific nomenclature when describing Osr1+ cells. Sometimes they only refer to Osr1-expressing cells as GFP+ cells, which makes the reader wonder whether Osr1+-expressing cells or traced cells derived from Osr1+ cells are meant (both are GFP-positive although the GFP is membrane bound in case of Cre-dependent reporter gene activation). This can be easily fixed.

REPLY: thank you for this notion. We have attempted to clarify this in the manuscript. For the sake of correctness, however, we kept the terms GFP+ or GFP- when this marker was used for e.g. comparative expression analysis. Also, when referring to the lineage tracing experiment, we have now used "mGFP" throughout to discriminate this from the GFP marker expressed from the Osr1^{GCE} allele.

Reviewer #2 (Remarks to the Author):

The authors have done an excellent job answering my previous comments. In particular, I am impressed with the careful comparative analysis of Osr1+ cells with other markers of muscle connective tissue fibroblasts - Pdgfra, Tcf4, and FAPs (as identified by Joe et al. FACS strategy). I also think the Osr1+/- co-culture with myogenic cells and Osr1+/- matrix with myogenic cells are an excellent addition to the paper. Overall, I think this paper is a strong contribution to the field and will be much appreciated.

I do have a few comments to improve the manuscript:

1. Line 35 Abstract: "developmental limb MCT cells expressing the transcription factor Osr1 give rise to adult FAPs". Since their lineage data shows that Osr1 lineage gives rise to a small subset of adult FAPs, I think this sentence should be modified to "developmental limb MCT cells expressing the transcription factor Osr1 give rise to a subset of adult FAPs".

REPLY: thank you; we have added this.

2. Line 42 Abstract: "We conclude that embryonic FAPs provide a pro-myogenic niche required for myogenesis in developing limbs..." That have not demonstrated that Osr1+ embryonic FAPs are required. Loss of Osr1 has relatively minor affects on muscle patterning and they have not done the experiment deleting Osr1+ cells (e.g. using Osr1^{GCE};RosaDTA mice). It would be more accurate to state "We conclude that Osr1+ embryonic FAPs provide a pro-myogenic niche that regulates myogenesis in developing limbs..."

REPLY: We had to shorten the abstract to 150 words upon editorial request, this sentence was deleted.

3. Line 24 Introduction: “muscle stem cells (satellite cells)” change to “muscle stem cells (termed satellite cells in the adult)”. Satellite cells are a term reserved for adult (older than 6 weeks) muscle stem cells which reside under the basement membrane of myofibers. The term “satellite cells” is not used for embryonic, fetal, or neonatal muscle stem cells.

REPLY: thank you, changed accordingly.

4. Line 54 Introduction: “It is widely accepted that myogenic cells do not contain intrinsic information that govern their place and time of differentiation” references

Ordahl and Dourain 1992

Aoyama and Asamoto 1988

Lance-jones Jones DB 408-419 1988

Lance-jones Jones DB 394-407 1988

This sentence is only true for vertebrate muscle. Please change sentence to read “It is widely accepted that vertebrate myogenic cells do not contain intrinsic information that govern their place and time of differentiation”. In fact, fly muscle is intrinsically specified.

REPLY: changed accordingly.

Also, not all the references are the appropriate references to cite.

Please use

Ordahl and Dourain 1992

Lance-jones Jones DB 394-407 1988 (only this Lance-Jones reference is appropriate)

REPLY: changed accordingly.

5. Line 69-70 Introduction: “The pro-myogenic function ascribed to FAPs during adult muscle regeneration is an intriguing parallel to the function lateral plate-derived MCT progenitors play during embryonic myogenesis. Whether or not these two cell types are related and what molecular mechanisms contribute to their pro-myogenic function is mostly unknown.” The relationship between adult and embryonic muscle connective tissue fibroblasts has been previously described in Mathew et al. 2011. At the time of the Mathew publication, FAPs had just been described and so this paper did not explicitly compare embryonic fibroblasts with adult FAPs. However, the paper clearly described (as shown by Tcf4 lineage experiments) that the lateral plate is the source of embryonic and adult muscle connective tissue fibroblasts. This paper should be cited here and in the Discussion in the paragraph beginning on Line 475.

REPLY: We have amended this issue by changing the sentence Line 61/62 “Muscle interstitial cells comprise connective tissue fibroblasts” to “Muscle interstitial cells comprise Tcf4+ connective tissue fibroblasts that originate from lateral plate mesoderm (citing Mathew et al.)”; and the sentence Line 69/70 referred to by the reviewer from “Whether or these two cell types are related” to “Whether or not FAPs and developmental MCT progenitors are related”.

6. Line 182-184 Results: “Intriguingly, GFP+ cells were equally distributed between Sca1₋ and Sca1₊ MCT cells (Supplementary Fig. 6a). This suggests that Osr1₊ cells gain Sca1 expression over time”. A similar conclusion is stated in Lines 47-478 Discussion: “An interesting notion is that adult Sca1₊ FAPs

originate from developmental Osr1⁺/Sca1⁻ cells". I am unclear whether the authors have the data to support the statement that Osr1⁺Sca1⁻ embryonic cells give rise to Sca1⁺ FAPs. The data needed would be to analyze adult Osr1^{GCE/+};RosatdTom mice given tam at E11.5 (when Osr1⁺ cells are confirmed to be Sca1⁻) via FACS and determine whether Tom⁺ cells ended up in FAPs gate and so are Sca1⁺. The Osr1⁺ Sca1⁺ cells in adults could be due to the expansion of a small de novo population of Osr1⁺Sca1⁺ cells. As far as I can tell from the data presented, they do not have any lineage data to support that Osr1⁺Sca1⁻ cells subsequently become Osr1⁺Sca1⁺. They do not necessarily have to do this lineage experiment, but if they do not have such data they should be careful in their interpretation and conclusions.

REPLY: we agree with the reviewer. We show that embryonic (E11.5 and E13.5) Osr1⁺ cells, which at these stages are Sca1⁻, give rise to interstitial PDGFRalpha cells at E18.5 (Fig. 3c), and to adult PDGFRalpha FAPs (Supplementary Fig. 4b). We however did not perform lineage analysis to assess and quantify the contribution of these Sca1⁻ cells to the Sca1⁺ pool. We changed the sentence "This suggests that Osr1⁺ cells gain Sca1 expression over time" to "It is therefore possible that Osr1⁺ cells gain Sca1 expression over time". We have removed the following sentences from the discussion: Lines 477-481 ("An interesting notion is that adult Sca1⁺ FAPs originate from developmental Osr1⁺/Sca1⁻ cells. Of note, Sca1⁻ cells still exist in the adult muscle interstitium referred to as osteo-chondrogenic progenitors. However these cells do not originate from developmental Osr1⁺/Sca1⁻ cells (Fig. 4c) but likely from a separate, Osr1⁻/Sca1⁻ population, which is in line with the anti-chondrogenic capacity of Osr1⁻").

7. Line 205-207 Results: "Osr1 progeny was observed exclusively in FAPs, while no lineage progeny was found in satellite cells or osteo-chondrogenic progenitors (Fig. 4c) showing that Osr1⁺ progenitors only contribute to the FAP pool." It is important that they explicitly mention that only 7% of the FAPs were derived from Osr1⁺ cells genetically labeled at P0. Thus Osr1⁺ cells are unlikely to be the only developmental source of FAPs.

REPLY: We think that we appropriately addressed this in the text. The last sentence of this paragraph reads "Altogether, this establishes that adult FAPs derive, at least in part, from developmental Osr1⁺ cells." Of note, the 7% contribution was assessed by FACS using whole limb muscle cell extracts. This means that also FAPs from muscles that were Osr1 negative during development were extracted. This is opposed by quantitation on tissue sections restricted to muscles that expressed Osr1 in development, which shows a higher contribution rate. Moreover it has to be taken into consideration that Tamoxifen pulsing was performed for two consecutive days making an absolute quantification impossible. This strategy was chosen to allow a clear statement on the fate of developmental Osr1⁺ cells (at a time when Osr1 is expressed in interstitial cells) to adult FAPs that cannot be confounded by expression of Osr1 in other cell types. Using a constitutive Osr1-Cre line would indeed likely lead to widespread unspecific labeling due to the expression of Osr1 in the early mesoderm at E7.5 and E8.5 (Wang et al. 2005 Dev Biol 288). We nevertheless recognize that it is an important point to state that FAPs obviously do not derive exclusively from Osr1⁺ progenitors. We have added the following sentence to the discussion: "Together, our data show that adult FAPs derive in part from Osr1⁺ progenitors, but this is unlikely the only developmental source."

8. Line 326 Results: "...we conclude Osr1 is a key transcription factor that maintains embryonic FAP cell identity." Given that Osr1 is not expressed in all embryonic muscle connective tissue fibroblasts, this statement only applies to the subset of fibroblasts that express Osr1. Please modify this sentence.

REPLY: We changed the sentence "...we conclude Osr1 is a key transcription factor that maintains embryonic FAP cell identity." to "...we conclude Osr1 is a key transcription factor that maintains embryonic Osr1+ FAP cell identity."

9. Lines 372-375 Results: "ECM produced by Osr1GCE/+ cells significantly enhanced myogenic cell proliferation in comparison to cultivation on Matrigel. ECM produced by Osr1 deficient cells (Osr1GCE/GCE) failed to promote myoblast proliferation in comparison to Matrigel (Fig. 9c)." Fig 9c shows that Ki67+Pax7+/Pax7+ cells when cultured on ECM produced by Osr1GCE/+ and Osr1GCE/GCE. However, the Fig does not compare these numbers when cultured just on Matrigel.

REPLY: thank you very much for spotting this mistake, we apologize. This has been corrected.

10. The title should be modified: "Odd skipped-related 1 (Osr1) identifies a population of embryonic fibro-adipogenic progenitors (FAPs) and regulates a pro-myogenic transcriptional program during limb development".

REPLY: Thank you for the suggestion. Upon editorial request we had to shorten the title to 15 words. The new title is: "Odd skipped-related 1 identifies a population of embryonic fibro-adipogenic progenitors regulating myogenesis during limb development".